# Evaluating Future Streamflow Patterns under SSP245 Scenarios: Insights from CMIP6

Kashif Haleem [1], Afed Ullah Khan [1,2,*], Jehanzeb Khan [3], Abdulnoor A. J. Ghanim [4] and Ahmed M. Al-Areeq [5]

1 National Institute of Urban Infrastructure Planning, University of Engineering and Technology, Peshawar 25000, Pakistan; kashifhaleem.niuip@uetpeshawar.edu.pk
2 Department of Civil Engineering, University of Engineering and Technology Peshawar, Bannu Campus, Bannu 28100, Pakistan
3 Higher Education Department, Government Post Graduate College Kohat, KPK, Kohat 26000, Pakistan; jehan8bio@yahoo.com
4 Civil Engineering Department, College of Engineering, Najran University, Najran 61441, Saudi Arabia; aaghanim@nu.edu.sa
5 Interdisciplinary Research Center for Membranes and Water Security, King Fahd University of Petroleum & Minerals (KFUPM), Dhahran 31261, Saudi Arabia; ahmed.areeq@kfupm.edu.sa
* Correspondence: afedullah@uetpeshawar.edu.pk; Tel.: +92-033-5854-7076

**Abstract:** The potential impacts of climate change on water resources in the Upper Indus Basin of Pakistan, a region heavily reliant on these resources for irrigated agriculture. We employ state-of-the-art global climate models from the CMIP6 project under the SSP245 scenario to evaluate changes in river runoff using the Soil and Water Assessment Tool (SWAT). Our findings indicate that temperature fluctuations play a crucial role in streamflow dynamics, given that the primary sources of river runoff in the Upper Indus Basin are snow and glacier melting. We project a substantial increase of approximately 18% in both minimum and maximum temperatures, precipitation pattern increases of 13–17%, and a significant rise in streamflow by 19–30% in the future, driven by warmer temperatures. Importantly, our analysis reveals season-specific impacts of temperature, precipitation, and streamflow, with increasing variability in projected annual changes as we progress into the mid and late 21st century. To address these changes, our findings suggest the need for integrated strategies and action plans encompassing hydroelectricity generation, irrigation, flood prevention, and reservoir storage to ensure effective water resource management in the region.

**Keywords:** climate change; CMIP6 models; SWAT modeling

## 1. Introduction

Global climate change is having a significant impact on water distribution networks in many regions of the globe. Watershed management failures, such as extensive deforestation, poor land use management, and unsustainable agricultural methods, are responsible for the worst scenarios [1]. Currently, environmental circumstances are very worrisome and need appropriate watershed policies for long-term sustainability. All climatic processes are unquestionably increasing in intensity [2,3]. Floods, warmer temperatures, and droughts are illustrations of extreme events that show the situation's severity. According to climate models, global temperatures will rise due to an increase in mean near-surface air temperature [4]. This may significantly improve the hydrological cycle variability, including variations in the precipitation, evapotranspiration, and flow rate. Freshwater resources are enormously important for human civilization and ecosystems, yet they are susceptible and may be damaged by global climate change [5–7]. In addition, the huge contradiction between the supply and demand of freshwater resources provides a significant need to forecast hydrological responses to climate change and to manage water resources appropriately.

Presently, one of the most pressing issues in the field of water quantity management is climate change. Keeping the acknowledgments of the Fifth Assessment Report of the Intergovernmental Panel on Climate Change in 2014 (IPCC AR5, 2014), temperatures might elevate by 3.7 °C by 2100. Global warming is triggering changes to watershed management across the globe. Thus, the impacts on water resources due to climate change are of greater concern for future water management and to respond effectively to the worst scenarios [8,9]. Many researchers such as [10–12] have used IPCC AR5 for the future projection and evaluation of future climate change [13]. Generally, global circulation model (GCM) is used for the climate change scenarios in many regions of the world. In addition, statistical or dynamic downscaling methods are essential for the evaluation of GCMs because of their high spatial resolutions which need proper assessment [14,15]. The Coupled Model Inter-Comparison Project (CMIP) was initiated in 1995 under the World Climate Research Program (WCRP) to assess the changes in climate scenarios through a multi-model context from the past to the future [13]. CMIP6 is the latest edition of CMIP, in which climatic projections are based on shared socioeconomic paths (SSP). Broadly, SSPs are updated and revised versions of the representative concentration pathways (RCPs), in which anthropogenic drivers are further improved for the assessment of future scenarios along with socio-economic development [16]. Furthermore, evaluation of the updated and revised phase of CMIP is necessary to make proper sustainable resources to challenge worse scenarios.

Inter-model comparison Project Phase 5 (CMIP5) models have greatly accompanied in determining climate change scenarios [17]. Previous researchers such as [18,19] used CMIP5 models to predict future drought risk and water availability. However, CMIP5-based GCMs reveal substantial uncertainty in future summer monsoon precipitation projections [20]. Numerous CMIP5-based GCMs failed in South Asia to capture monsoon precipitation; at the very outset of monsoon and large-scale variability in the future climate [21,22]. In CMIP6, the main aims are the advancement and improvement of the parametrization and representation of the climate change projections. Recent researchers such as [23,24] have stated that the assessment of the CMIP6-based general circulation model (GCM) apprehended efficient results as compared to CMIP5-based GCMs during the evaluation of the Indian summer monsoon.

A combination of the representative concentration pathway (RCP) and alternative techniques has been offered in the CMIP6 climate model for modeling future emission scenarios. A number of innovative combinations of SSP scenarios have been developed for use in CMIP6. These scenarios are based on updated versions of the SSP scenarios from CMIP5 (SSP119, SSP370, SSP434, SSP245, and SSP585). SSP scenarios encompass socioeconomic elements such as population growth, economy, urbanization, and other factors [13]. Wider equilibrium climate sensitivity with an expanding temperature range of 1.5–4.5 °C is one of the improvements to CMIP6 scenarios. CMIP6 models were projected to enhance capabilities and minimize uncertainty over the earlier CMIP5 and CMIP3 models [25]. SSP245 and SSP585 predict that by the end of the 21st century, radiative forcing will have stabilized at 4.5 and 8.5 W m$^{-2}$, respectively. The SSP245 scenario is subjective for most countries pursuing sustainable growth. The SSP585 scenario, on the other hand, emphasizes the worst-case scenario (a fossil fuel-based economy) as well as the repercussions of unconventional energy development. [26].

Currently, various tools are available to evaluate the impacts on watersheds mainly due to alterations in river runoff and base flow. However, most of the hydrologic methods have similarities within the watersheds, corresponding catchments, and hydrological modeling [27]. Among all, the most appealing and conceptual approach for the evaluation of watersheds is to use the Soil and Water assessment tool (SWAT) model [28]. Moreover, the SWAT model is widely used around the globe for the evaluation of hydrological processes and also for environmental and ecological variations at any catchment scale [29]. The SWAT model allows interconnection among variant physical processes [30].

The Upper Indus Basin of Pakistan supplies water to one of the world's largest irrigation systems. For the past several decades, the region has been experiencing imminent

threats from climate change. The runoff in the Upper Indus Basin depends on a host of factors, including the melting of seasonal snowpack, glacier melting, and precipitation [31]. The erratic changes in maximum and mean temperature during summer and especially in winter have intensified the glacier melting and consequently river flow, adding to the woes of the indigenous community living in the Upper Indus Basin [32]. Mountain ecosystems are widely acknowledged to be the most sensitive to climate change [33]. These vulnerabilities are anticipated to be exacerbated by disproportionate warming in mountain areas, notably in Gilgit Baltistan, which is one of the world's most mountainous and glaciated countries outside of the Polar Regions. This study mainly focuses on using CMIP6-based GCMs for the projection of future climate change and their impacts on river runoff using the SWAT model. The main objectives of this research include: utilizing CMIP6-based Global Climate Models (GCMs) within the Soil and Water Assessment Tool (SWAT) for streamflow projection; investigating the impacts of climate change on streamflow; and assessing the overall changes in projected precipitation, as well as maximum and minimum temperatures, and their effects on streamflow in the upper reaches of the Indus Basin.

## 2. Study Area Description, Data Collection and Methods

### 2.1. Study Area Description

The study area is chiefly focused on Gilgit region, which is the upper region of the Indus River. Indus River Basin is communal among four countries such as Afghanistan, China, India, and Pakistan with a total area of $1.1 \times 10^6$ Km$^2$ [34] and it is extended between 32.48° to 37.07° N and 67.33° to 81.83° E [35]. Gilgit Baltistan's districts are particularly vulnerable to climate change and have very different terrain and geology. Every year, communities in these areas face a variety of environmental threats. District Ghizer is situated in the far northwestern portion of Gilgit Baltistan, bordering Afghanistan via the Wakhan strip to the northwest and China to the north. It is rich in cultural, linguistic, and environmental diversity. The region also hosts one of the largest pools of flora and fauna adding to its diversity. However, climate change has been causing several issues for the region, such as water insecurity, changes in land cover, displacement of population, and loss of natural habitat for the last few decades [36]. Moreover, the warming in the upper regions of Indus Basin tends to be from 0.3 °C to 0.7 °C higher in future even if the level of global warming is kept to 1.5 °C raising the vulnerability of the region to its worst [37]. The seasonal snow cover varies from 95% in winter to 5% in summer [38]. The cryosphere is the major land type, with a sparse cover of plants. Intensive farming is often practiced across the area during agricultural operations. Gilgit Baltistan Basin area is 45,400 km$^2$ and its elevation ranges from 547 m (lowest) to 8060 m (highest) as shown in Figure 1.

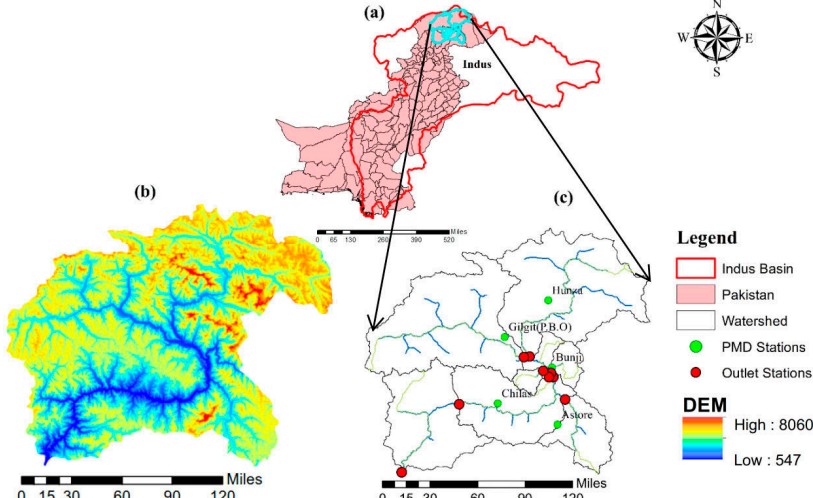

**Figure 1.** (**a**) Pakistan and Indus Basin, (**b**) digital elevation model (DEM), and (**c**) watershed of the upper regions of the Indus Basin.

*2.2. Data Collection*

SWAT model is a data-intensive model that requires inputs such as DEM, land use, soil map, and hydroclimatic data. National Aeronautics and Space Administration (NASA) provided the Advanced Spaceborne Thermal Emission and Reflection Radiometer (ASTER) Global Digital Elevation Model. The ASTER GDEM has a resolution of 30 m × 30 m. Figure 1b shows how digital elevation model (DEM) data are utilized in the SWAT model to delineate watersheds. To locate hydrologic response units (HRUs), the SWAT model employs a land use and soil map. The United States Geological Survey (USGS) provided land use data from the Moderate Resolution Imaging Spectroradiometer (MODIS) with a spatial resolution of 500 m. In addition, the Food and Agriculture Organization (FAO) provided a worldwide soil map. Figures 2 and 3 show the land use and soil types in further detail. Furthermore, for its analysis, SWAT requires daily precipitation and temperature (minimum and maximum), for their simulations. For the years 1982 to 2013, daily meteorological data for study area (Table 1) were obtained from the Pakistan Meteorological Department. Due to the scarcity of consistent, long-term meteorological data in the study area, we were forced to use only five stations. A broader network of stations would provide a more comprehensive understanding of precipitation and temperature variability. This study attempted to address temperature effects in the higher parts of the basin, so the model's sensitivity may not fully represent the entire basin due to this constraint. The solar radiation, humidity, and wind speed data were incorporated in SWAT model from Climate Forecast System Reanalysis (CFSR) source. A weather generator module integrated with SWAT generates daily humidity, solar radiation, and wind speed data based on precipitation and temperature. However, solar radiation, humidity, and wind speed data are not directly produced by this module. Rather, it depends on these parameters as functions of precipitation and temperature. Furthermore, hydrological data are necessary for completing an uncertainty analysis following a SWAT model simulation in order to compare the model's results to observed data. For the years 1982 to 2013, hydrological data (discharge monthly data) were obtained for Doyian station at Astore River from Pakistan Water and Power Development Authority.

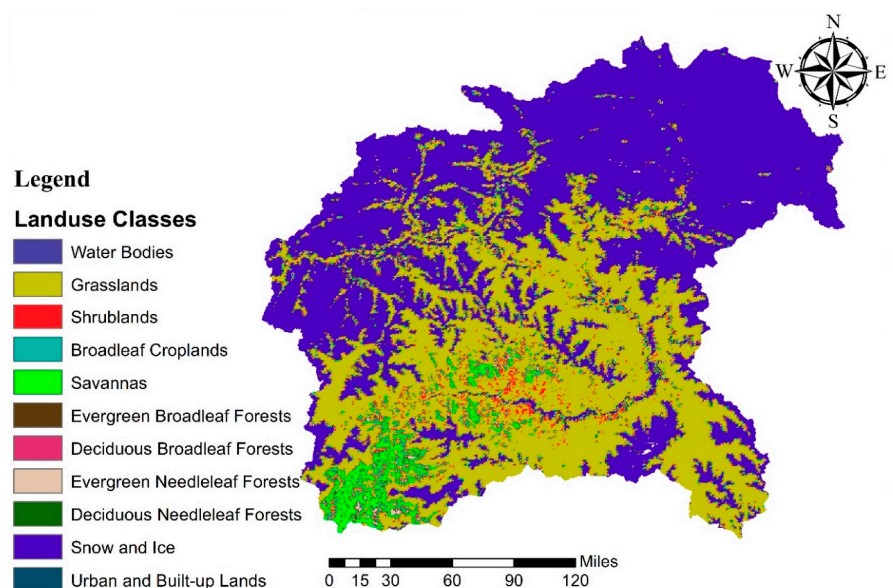

**Figure 2.** Distribution of various land use classes in the upper regions of Indus Basin.

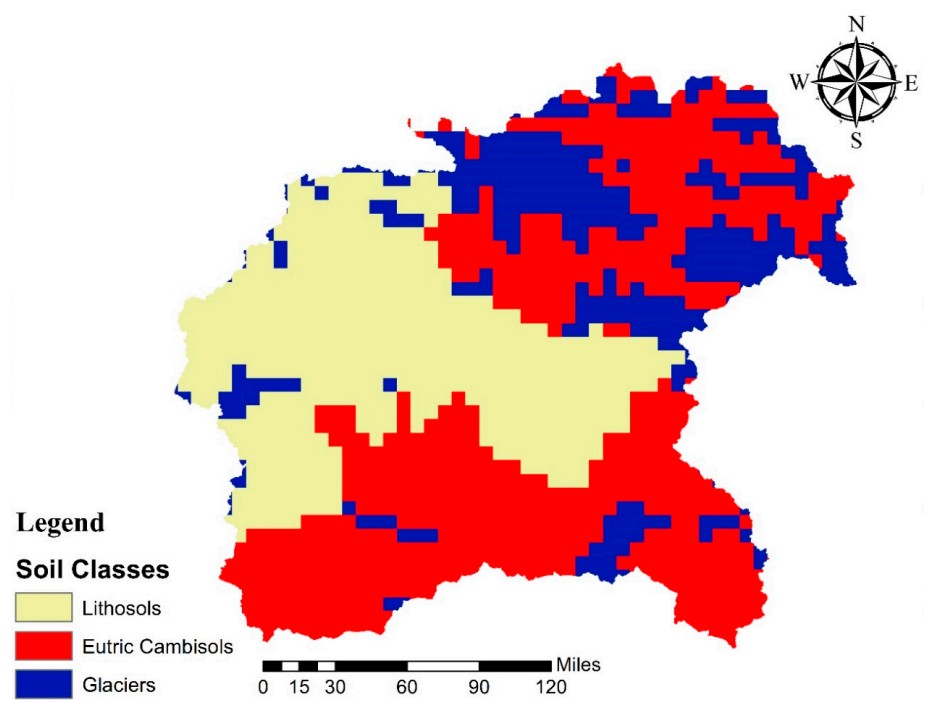

**Figure 3.** FAO soil classes in the upper regions of Indus Basin.

**Table 1.** Meteorological station in the study area.

| Station Name | Latitude | Longitude | Elevation (m) |
|:---:|:---:|:---:|:---:|
| Bunji | 35.6667 | 74.6333 | 1372 |
| Gilgit (P.B.O) | 35.9167 | 74.3333 | 1460 |
| Chilas | 35.4167 | 74.1 | 1250 |
| Hunza | 36.3167 | 74.65 | 2156 |
| Astore | 35.3333 | 74.9 | 2168 |

*2.3. Methodology*

2.3.1. Bias Correction of GCMs

Bias correction procedures use a transformation algorithm to change the performance of climate models. Biases between historical climate variables that are observed and those that are simulated can be used to set parameters for a bias correction algorithm. This algorithm is used to correct simulated historical climate data that are not accurate. Since the correction algorithm and its parameters for current climate conditions are thought to be valid for future climate conditions as well, they are thought of as "stationary". Thus, the same correction method is used to correct future climate data. However, how well a bias correction algorithm operates under conditions other than those used for parametrization is uncertain. A good performance during the evaluation period does not mean that the same thing will happen in the future. Teutschbein and Seibert [39] produced a comprehensive explanation and suggested that a method that works well for the present is more likely to perform well in the future than a method that performs poorly for current scenarios.

Climate models are often compared to observed data using various approaches to reduce uncertainty if they show very little or no relation to observed data. GCMs are adjusted via transformation algorithms in bias correction approaches to relate GCM outputs to observable data. Moreover, the foremost aim of the biases is to correlate observed data with the simulated data and to follow the same trend with deep statistics for the future projections of the GCMs in that particular region [40]. Currently, climate model data for hydrologic modeling (CMhyd) are used to make climate models more credible. This tool is very easy to use in evaluating GCMs in any study area [41]. In addition, CMhyd tool includes biasing methods such as linear scaling (additive and multiplicative), temperature

variance scaling, precipitation power transformation, precipitation local intensity scaling, delta change correction (additive and multiplicative), and precipitation and temperature distribution mapping. Figure 4 illustrates a flowchart of the data processing and assessment in the CMhyd tool. In this study, linear scaling (additive and multiplicative) produced the best results for the evaluation of GCMs of all bias correction approaches, and the same strategy was used [42]. Overlapping daily observed data (baseline period from 1985 to 2013) is required for historical data correction under the transformation algorithm while using CMyd tool. Following the correction of historical data, the same procedure is used for the correction of future data.

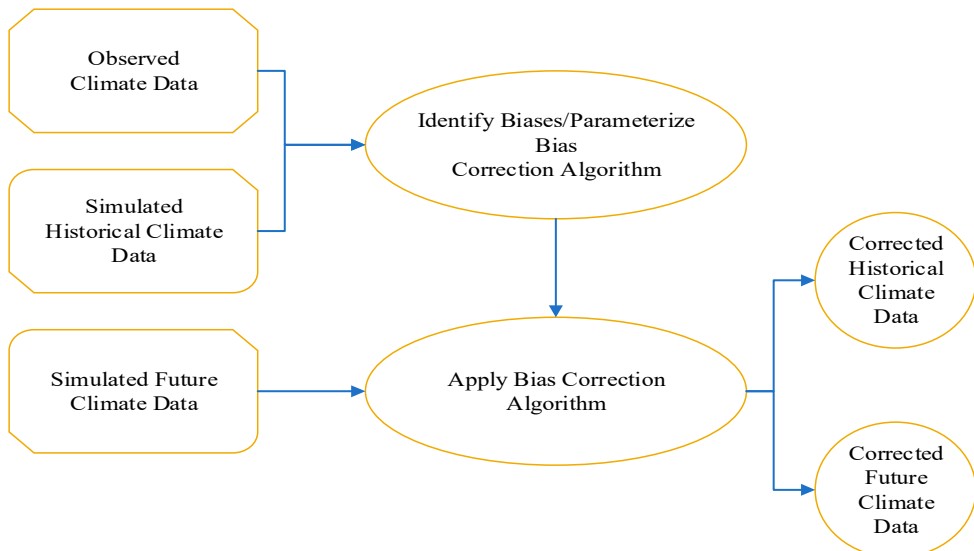

**Figure 4.** Depicts the overview of the bias correction method through CMhyd tool.

2.3.2. Description of SWAT Model

SWAT is a physical-based hydrological model developed by the United States Department of Agriculture in the 1990s. To examine the hydrology of any watershed, it works in collaboration with geographic information systems [43,44]. SWAT model requires DEM data for watershed delineation, land use and soil map for defining HRUs, and daily meteorological (precipitation and temperature) data for simulations [45,46]. For simulation, the SWAT model manages to make use of the water balance Equation (1). The flowchart shows the SWAT model process in Figure 5.

$$SW_t = SW_o + \sum_{i=o}^{t} (R_{day} - Q_{surf} - E_a - W_{seep} - Q_{gw})_i \tag{1}$$

whereas:

$SW_t$ = final soil moisture content, mm;
$SW_o$ = initial soil moisture content of the *i*-th day, mm;
$t$ = time, d; $R_{day}$ = precipitation of the *i*-th day, mm;
$Q_{surf}$ = surface runoff of day *i*, mm;
$E_a$ = amount of evapotranspiration on day *i*, mm;
$W_{seep}$ = value of seepage of water from the soil into deeper layers *i*, mm;
$Q_{gw}$ = return flow amount on day *i*, mm.

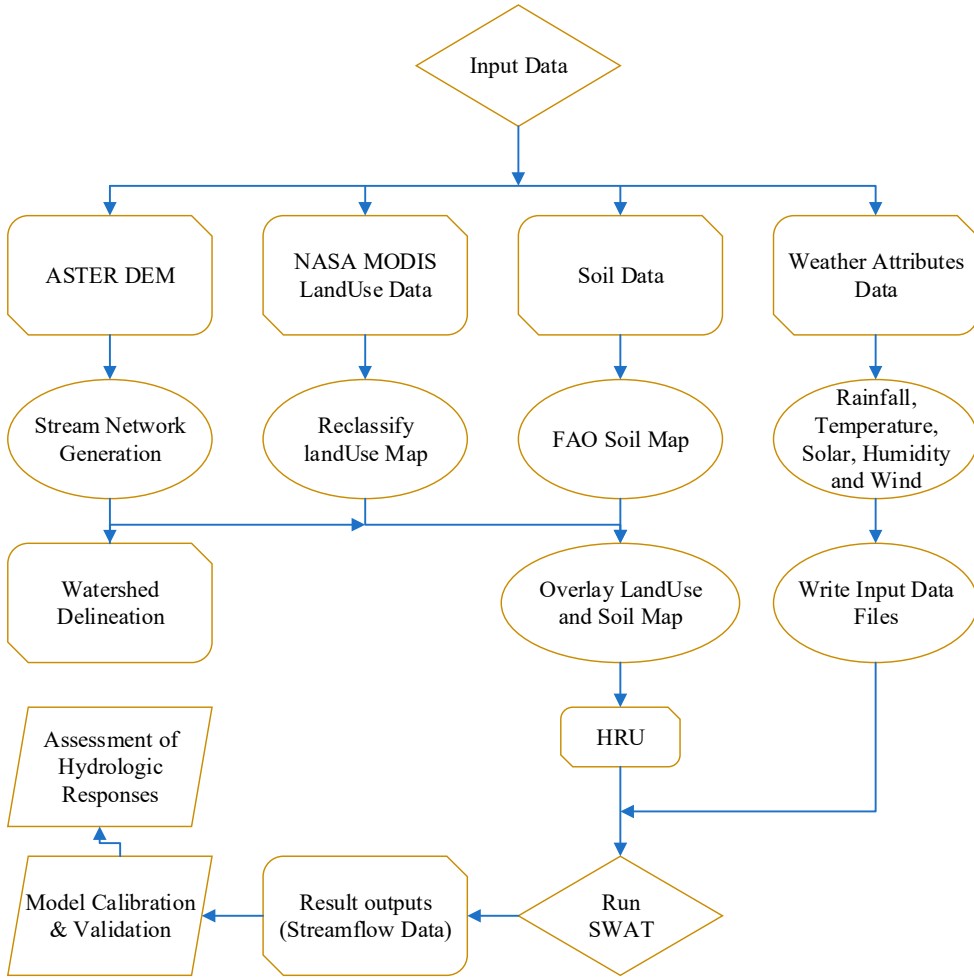

**Figure 5.** Processes followed in the application of the SWAT model.

### 2.3.3. SWAT Model Calibration and Validation

Usually, after SWAT modeling, the simulated results are evaluated in SWAT-CUP (SWAT Calibration and Uncertainty Procedures) to remove any uncertainty in the parameters and select the best fittings. Due to its suitability and efficiency in fitting the parameters, most researchers use SWAT-CUP automatic calibration with the Sequential Uncertainty Fitting program algorithm (SUFI-2) for uncertainty analysis [47]. SWAT-CUP has variant uncertainty models such as Generalized Likelihood Uncertainty Estimation (GLUE), Parameter Solution (ParaSol), Markov Chain Monte Carlo (MCMC), and Particle Swarm Optimization (PSO). For better results, the SWAT-CUP model uses 2/3 of the data for calibration and 1/3 of the data for validation. Pakistan is a data-scarce country. Long and consistent historical streamflow records are available at Doyian station, which is located at Astore River. Therefore, Doyian station was selected for model calibration. We utilized SWAT-CUP (SWAT Calibration and Uncertainty Procedures) with the Sequential Uncertainty Fitting program algorithm (SUFI-2) for this calibration process. SWAT-CUP model is calibrated using monthly data from 1985 to 1999 and validated from 2000 to 2010. In addition, calibration is a technique for establishing the best fit between observed and simulated data in order to obtain the best values for representative functions. Statistical analysis methods that are most typically utilized are Nash–Sutcliffe efficiency (*NSE*) [48], coefficient of determination ($R^2$) [49], percent bias (*PBIAS*) [50], and root mean square error to the standard deviation ratio (*RSR*) [51] for evaluating the climatic data sets, as well as for evaluating the performance of the model. *NSE* in Equation (2) is a standardized statistical indicator having a range of values from 0 to 1. Close to 1 value implies a suitable

fit between the modeled and the observed data. $R^2$ in Equation (3) shows the ability of the model in comparison with observed data. *PBIAS* in Equation (4) estimates the model values, which may be lower (underestimated) or higher (overestimated), values close to zero are the most accurate model simulation, and *RSR* in Equation (5) is another statistical indicator for model performance in which the most optimum value is zero. The model efficiency is always tested in the validation period in which the calibrated parameters remain fixed, and changes are made in the input data to assess the representative functions; their ranges are given in Table 2.

$$NSE = 1 - \frac{\sum_{i=1}^{N}(X_i - Y_i)^2}{\sum_{i=1}^{N}\left(X_i - \overline{X}\right)^2} \tag{2}$$

$$R^2 = \left[\frac{\sum_{i=1}^{N}(X_i - \overline{X})(Y_i - \overline{Y})}{\sqrt{\sum_{i=1}^{N}(X_i - \overline{X})^2}\sqrt{\sum_{i=1}^{N}(Y_i - \overline{Y})^2}}\right]^2 \tag{3}$$

$$PBIAS = \frac{\sum_{i=1}^{N}(Y_i - X_i)}{\sum_{i=1}^{N}X_i} \times 100 \tag{4}$$

$$RSR = \left[\frac{\sqrt{\sum_{i=1}^{N}(X_i - Y_i)^2}}{\sqrt{\sum_{i=1}^{N}(X_i - \overline{X})^2}}\right] \tag{5}$$

where $X$ and $Y$ are the modeled and observed streamflow, $\overline{X}$ and $\overline{Y}$ are the mean modeled and observed streamflow, and $N$ is the pairs of data [52].

**Table 2.** Statistical indicators for model performance evaluation [53].

| Performance Rating | NSE | $R^2$ | RSR | PBIAS (%) |
|---|---|---|---|---|
| Very good | $0.75 < NSE \leq 1$ | $0.75 < R^2 \leq 1$ | $0 \leq RSR \leq 0.5$ | $-10 < PBIAS < 10$ |
| Good | $0.65 < NSE \leq 0.75$ | $0.65 < R^2 \leq 0.75$ | $0.5 < RSR \leq 0.6$ | $\pm 10 \leq PBIAS < \pm 15$ |
| Satisfactory | $0.5 < NSE \leq 0.65$ | $0.5 < R^2 \leq 0.65$ | $0.6 < RSR \leq 0.7$ | $\pm 15 \leq PBIAS < \pm 25$ |
| Unsatisfactory | $NSE \leq 0.5$ | $R^2 \leq 0.5$ | $RSR > 0.7$ | $PBIAS \geq 25$ |

### 2.3.4. Climate Change Forecasting

In climate change scenarios, uncertainty arises due to limitations such as the selection of future climate models, insufficient physical understanding of various self-connections, and computational capabilities [54]. Efficient corrections are needed in order to make good choices in various climate change situations. In this study, various GCMs were utilized for the projection of future climate change, which were downloaded from World Climate Research Programme webpage (https://www.wcrp-climate.org/ (accessed on 5 January 2023)). The biased corrected GCMs data were divided into four sections: (1974–1993), (1994–2013), (2014–2033), and (2074–2094). The description of each CMIP6-GCM is described in Table 3.

**Table 3.** Overview of the CMIP6-GCMs adopted for this study.

| Modeling Center | Model Institute | ID | Resolution (km) |
|---|---|---|---|
| Commonwealth Scientific and Industrial Research Organization, Australian Research Council Centre of Excellence for Climate System Science, Australia | ACCESS-CM2 | ACCESS | $192 \times 144$ |
| Centro Euro-Mediterraneo per i Cambiamenti, Italy | CMCC-CM2-SR5 | CMCC | $288 \times 192$ |
| Geophysical Fluid Dynamics Laboratory, United States | GFDL-ESM4 | GFDL | $288 \times 180$ |
| Geophysical Fluid Dynamics Laboratory, United States | GFDL-CM4 | GFDL | $360 \times 180$ |
| Institute for Numerical Mathematics, Russia | INM-CM4-8 | INM | $180 \times 120$ |

**Table 3.** *Cont.*

| Modeling Center | Model Institute | ID | Resolution (km) |
|---|---|---|---|
| Institute for Numerical Mathematics, Russia | INM-CM5-0 | INM | $180 \times 120$ |
| Norwegian Climate Centre, Norway | NorESM2-LM | NorESM2 | $144 \times 90$ |
| Norwegian Climate Centre, Norway | NorESM2-MM | NorESM2 | $288 \times 192$ |
| The Taiwan Earth System Model version 1 | TaiESM1 | TaiESM1 | $100 \times 138$ |
| Met Office Hadley Centre, United Kingdom | UKESM1-0-LL | UKESM1 | $192 \times 144$ |

## 3. Results and Discussion

### 3.1. Model Calibration and Validation

In this research, the SWAT model was used to conduct a comprehensive analysis based on observed daily precipitation and temperature data spanning from 1982 to 2013. We also incorporated observed mean monthly discharge data for the same time frame to evaluate the model's accuracy. A long and consistent historical streamflow record is available at Doyian station located at Astore River. Therefore, the Doyian station was selected as the focal point for model calibration. Monthly calibration (1985–1999) and validation (2000–2010) results are shown in Figure 6, a three-year warm-up phase (1982–1984) was utilized to establish the initial soil water conditions. Sensitivity analysis was required to identify the factors responsible for the good match between simulated and observed streamflow data. Table 4 describes the 16 most effective model-calibrated parameters. The model's accuracy in predicting watershed conditions was assessed using $R^2$, *NSE*, *PBIAS*, and *RSR*. Table 5 reveals that statistical indicators during the calibration phase are notably higher than those during the validation period. The model's performance may not be as accurate in the validation phase because it may not generalize well to new or unknown data when these calibrated parameters are checked using validation data, which represents different time periods. Monthly simulated results are statistically significant, confirming that the SWAT model can effectively simulate the hydrological processes in the study area. Figure 6 shows the observed and simulated monthly streamflow at the outlet station during the calibration and validation period. A visual comparison of the simulated and observed manual streamflow reveals that the model fits and performs well over the study area. In both the calibration and validation periods, the model shows a relatively high P-factor and an R-factor that is above 0.7, suggesting that the model performs well in simulating hydrological processes. For calibration (P-factor = 0.86, R-factor = 0.82) and for validation (P-factor = 0.85, R-factor = 0.71).

**Table 4.** Parameters sensitive to streamflow, their fitted values, and initial ranges.

| Parameter | Description | Fitted Value | Ranges |
|---|---|---|---|
| CN2.mgt | SCS runoff curve number. | 63.1 | (35, 98) |
| ALPHA_BF.gw | Baseflow alpha factor (days). | 0.83 | (0, 1) |
| GW_DELAY.gw | Groundwater delay (days). | 409.12 | (0, 500) |
| GWQMN.gw | Threshold depth of water in the shallow aquifer required for return flow to occur (mm). | 2118.01 | (0, 5000) |
| SURLAG.hru | Surface runoff lag time. | 12.52 | (0.05, 24) |
| SLSOIL.hru | Slope length for lateral subsurface flow. | 84.69 | (0, 150) |
| EPCO.hru | Plant uptake compensation factor. | 0.45 | (0, 1) |
| SOL_K(..).sol | Saturated hydraulic conductivity. | 666.74 | (0, 2000) |
| SFTMP.bsn | Snowfall temperature. | 7.12 | (−20, 20) |
| SMTMP.bsn | Snow melt base temperature. | 17.60 | (−20, 20) |
| SNO50COV.bsn | Snow water equivalent that corresponds to 50% snow cover. | 0.79 | (0, 1) |
| CH_K2.rte | Effective hydraulic conductivity in main channel alluvium. | 76.36 | (−0.01, 500) |
| ESCO.hru | Soil evaporation compensation factor. | 0.94 | (0, 1) |

**Table 4.** *Cont.*

| Parameter | Description | Fitted Value | Ranges |
|---|---|---|---|
| LAT_TTIME.hru | Lateral flow travel time. | 34.14 | (0, 180) |
| CNOP{...mgt | SCS runoff curve number for moisture condition. | 16.84 | (0, 100) |
| SNO_SUB.sub | Initial snow water content. | 92.63 | (0, 150) |

**Table 5.** Statistical indicators for model calibration and validation that govern the streamflow.

| Calibration (1985–1999) | | | | Validation (2000–2010) | | | |
|---|---|---|---|---|---|---|---|
| $R^2$ | NSE | PBIAS | RSR | $R^2$ | NSE | PBIAS | RSR |
| 0.82 | 0.77 | 13.5 | 0.48 | 0.71 | 0.68 | 14.2 | 0.57 |

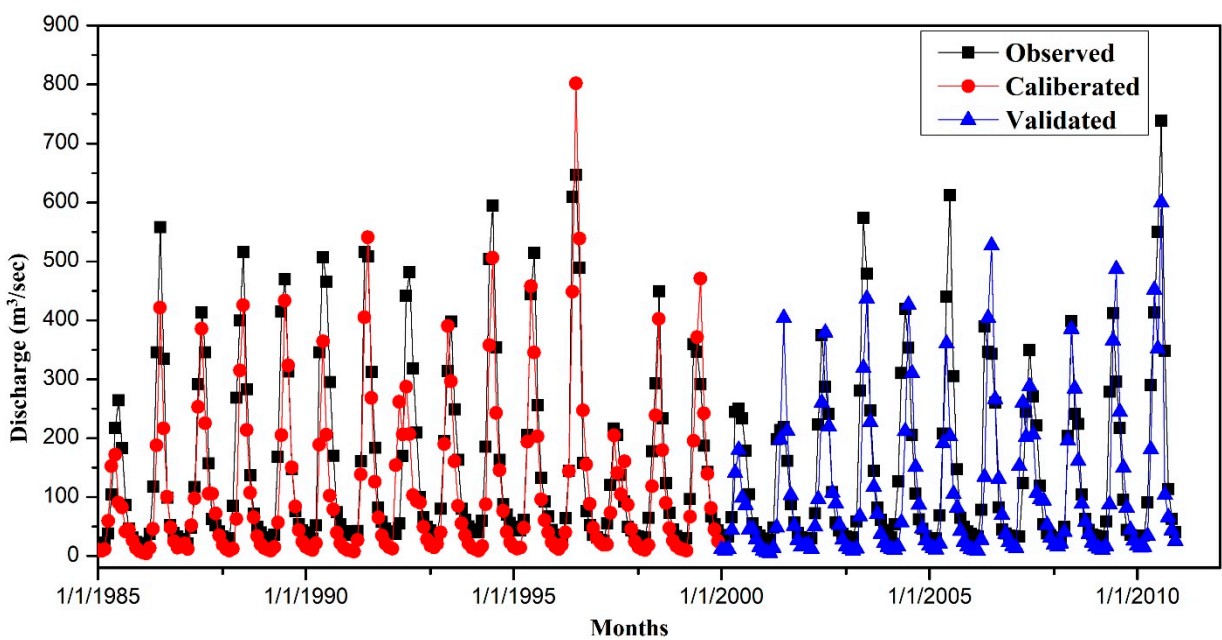

**Figure 6.** SWAT model streamflow simulation for calibration and validation period.

### 3.2. Temporal Patterns of Precipitation

In this study, a smoothing curve was used to identify trends or patterns in meteorological parameters. These curves draw insights from complex data and aid in decision-making. The Loess, local polynomial regression, was used to estimate and plot the smooth curve. The blue line in Figure 7 is actually the smooth line that represents a fitted or smoothed line that summarizes the relationship between two variables. The gray shaded area around the smooth line is the confidence interval. It is a graphical representation of the range of values within which the true population parameter is likely to fall with a certain level of confidence. The red data points that are located outside the gray shaded area fall outside the range of values that the model predicts with the given confidence level. This may indicate that these points are outliers or do not conform to the expected pattern as described by the smooth line.

Precipitation patterns behaved differently in each season, as seen in Figure 7, which displays the differences in precipitation in each season from 1985 to 2013. Spring and summer have shown a rising tendency throughout the course of the period, with the majority of the fluctuation occurring during the spring season. Winter and autumn seasons, on the other hand, have a plummeting trend throughout the period. The highest amount of mean monthly precipitation, 7.37 (mm), was recorded in the spring, while summer had the least, 3.55 (mm). In addition, Table 6 contains a summary statistic of mean seasonal

precipitation data for the period (1985–2013). Moreover, in Figure 8, the distribution of data points from various GCMs shows a high frequency of occurrence within the range of 0 to 2.5 (mm). High peaks of almost 12 (mm) can be seen in the NorESM2-LM model. Mean annual precipitation based on the SSP245 scenario is depicted in Table 7, which suggests that precipitation has a decreasing trend in the historical period, and then in the 2014–2033 and long-term (2074–2094) evaluation it seems that mean annual precipitation would increase, i.e., 1.57 (mm), which is 0.25 mm greater than the baseline period of 1.32 (mm).

**Table 6.** Summary statistics of average seasonal precipitation (mm) from 1985 to 2013.

| Statistics | Winter | Spring | Summer | Autumn |
|---|---|---|---|---|
| Min. | 0 | 0.01 | 0.07 | 0 |
| 1st Quartile | 0.42 | 1.14 | 0.495 | 0.165 |
| Median | 1.08 | 1.92 | 0.77 | 0.38 |
| Mean | 1.4 | 2.4 | 0.89 | 0.69 |
| 3rd Quartile | 1.96 | 3.31 | 1.14 | 0.64 |
| Max. | 5.6 | 7.37 | 3.55 | 5.74 |
| Standard Deviation | 0.65 | 1.08 | 0.5 | 0.29 |

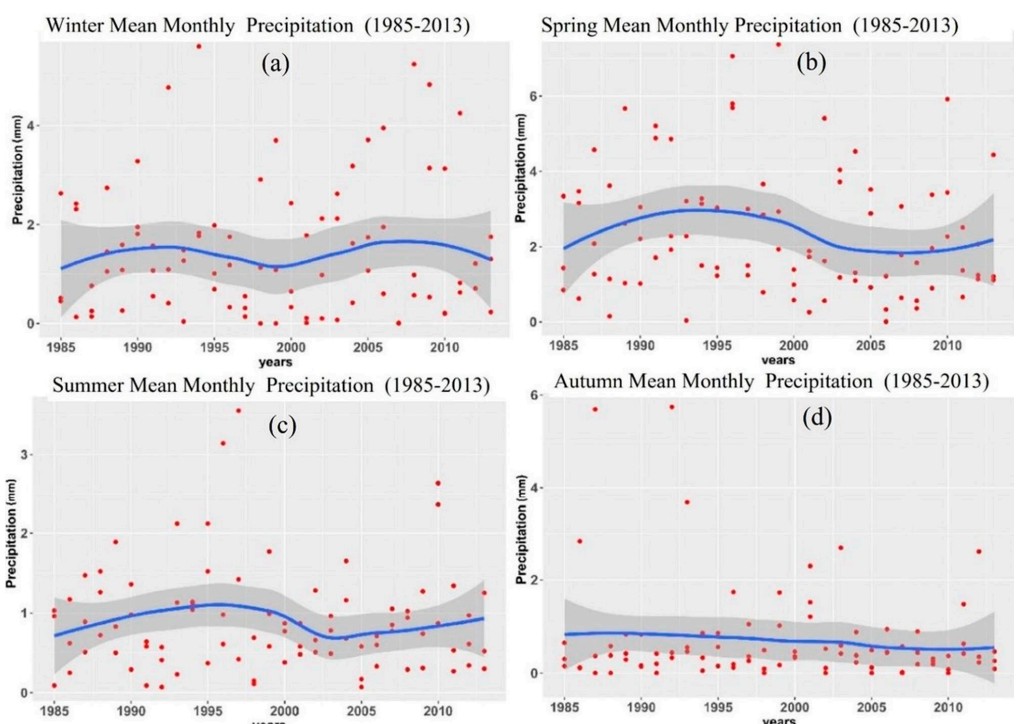

**Figure 7.** Mean monthly precipitation patterns over the period (1985–2013) for (**a**) Winter, (**b**) Spring, (**c**) Summer and (**d**) Autumn.

**Table 7.** Mean annual precipitation (mm) variations of baseline and GCMs under SSP245 scenario.

| Data Set | Historical 1974–1993 | Baseline 1994–2013 | 20's 2014–2033 | 80's 2074–2094 |
|---|---|---|---|---|
| Observed | - | 1.32 | - | - |
| ACCESS-CM2 | 1.39 (+4%) | 1.34 | 1.35 (+1%) | 1.57 (+17%) |
| CMCC-CM2-SR5 | 1.3 (2%) | 1.27 | 1.32 (+4%) | 1.44 (+13%) |
| GFDL-ESM4 | 1.32 (3%) | 1.28 | 1.31 (+2%) | 1.51 (+18%) |
| INM-CM4-8 | 1.35 (−4%) | 1.41 | 1.44 (+2%) | 1.57 (+11%) |
| INM-CM5-0 | 1.4 (+4%) | 1.35 | 1.38 (+2%) | 1.51 (+12%) |

**Table 7.** *Cont.*

| Data Set | Historical 1974–1993 | Baseline 1994–2013 | 20's 2014–2033 | 80's 2074–2094 |
|---|---|---|---|---|
| NorESM2-LM | 1.4 (3%) | 1.36 | 1.43 (+5%) | 1.51 (+11%) |
| NorESM2-MM | 1.38 (+7%) | 1.29 | 1.34 (+4%) | 1.51 (+17%) |
| TaiESM1 | 1.35 (−1%) | 1.37 | 1.39 (+1%) | 1.52 (+11%) |
| UKESM1-0-LL | 1.37 (+1%) | 1.35 | 1.38 (+2%) | 1.55 (+15%) |
| GFDL-CM4 | 1.4 (+4%) | 1.35 | 1.4 (+4%) | 1.53 (+13%) |

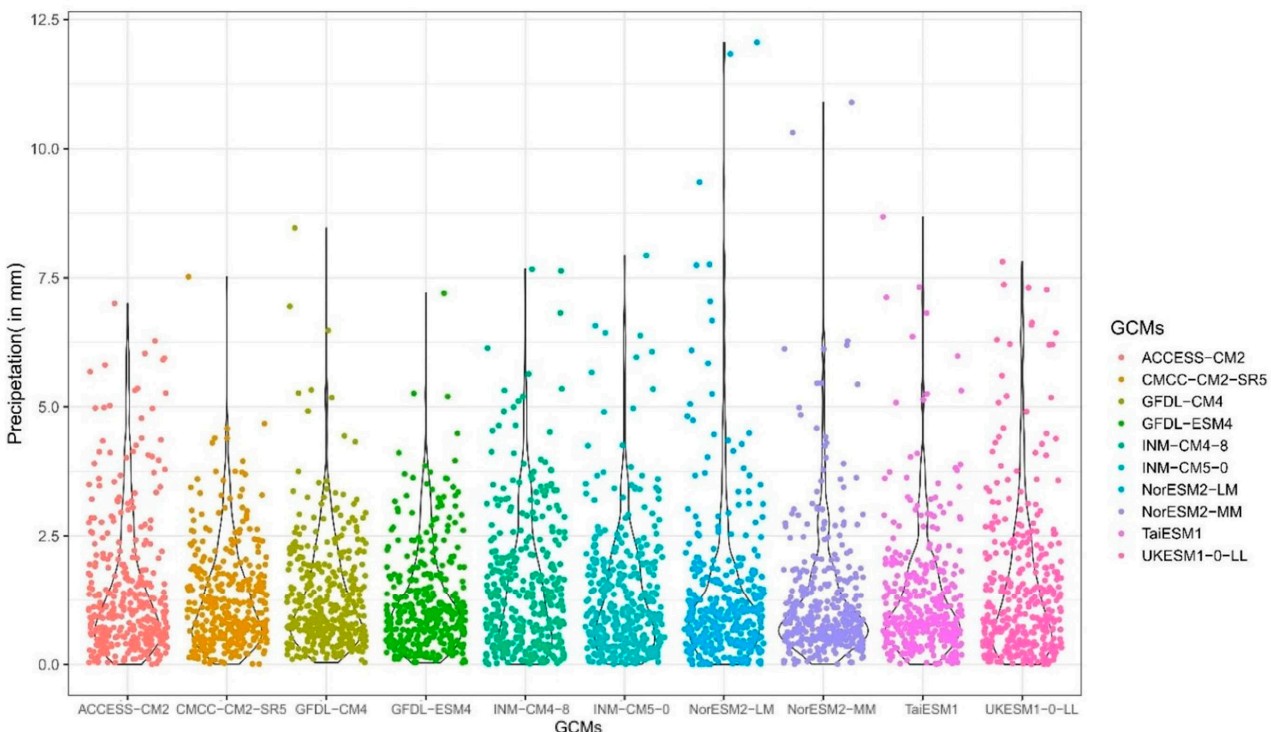

**Figure 8.** Mean monthly projected precipitation (mm) spread and variation of GCMs under SSP245 scenario from 1974 to 2094.

### 3.3. Temporal Patterns of Temperature

In our study, temperature exhibits significant variation throughout the 1985–2013 period in the upper regions of the Indus Basin. Specifically, mean monthly maximum and minimum temperatures reveal diverse trends, with a noticeable increase during the summer months but a sharp plunge during the winter season. Moreover, the mean maximum temperature in spring and autumn shows a steady decline, while the mean minimum temperature exhibits a consistent upward trend throughout the entire period. In Figure 9, summer temperatures tend to rise, whereas winter temperatures tend to fall. This phenomenon leads to increased glacier melting during the summer due to higher temperatures and amplified snowfall during the winter, subsequently contributing to heightened streamflow during the summer season. For further insight, Table 8 provides a summary of the temperature data for the period of 1985–2013, revealing that the highest temperature recorded was 39.7 °C, while the lowest temperature plummeted to −6.7 °C. It is essential to note that these temperature trends have future implications, as illustrated in Table 9 under the SSP245 scenario. This trend is critical, as it may exacerbate the challenges associated with watershed management and water resource sustainability in the Upper Indus Basin. Figure 10 further supports this observation, depicting the variance and dispersion of projected mean monthly temperature (both maximum and minimum) data. While the seasonal trends exhibit minimal variation, there is a progressive rise in the total anticipated temperature, reaching almost 18% in the Upper Indus Basin.

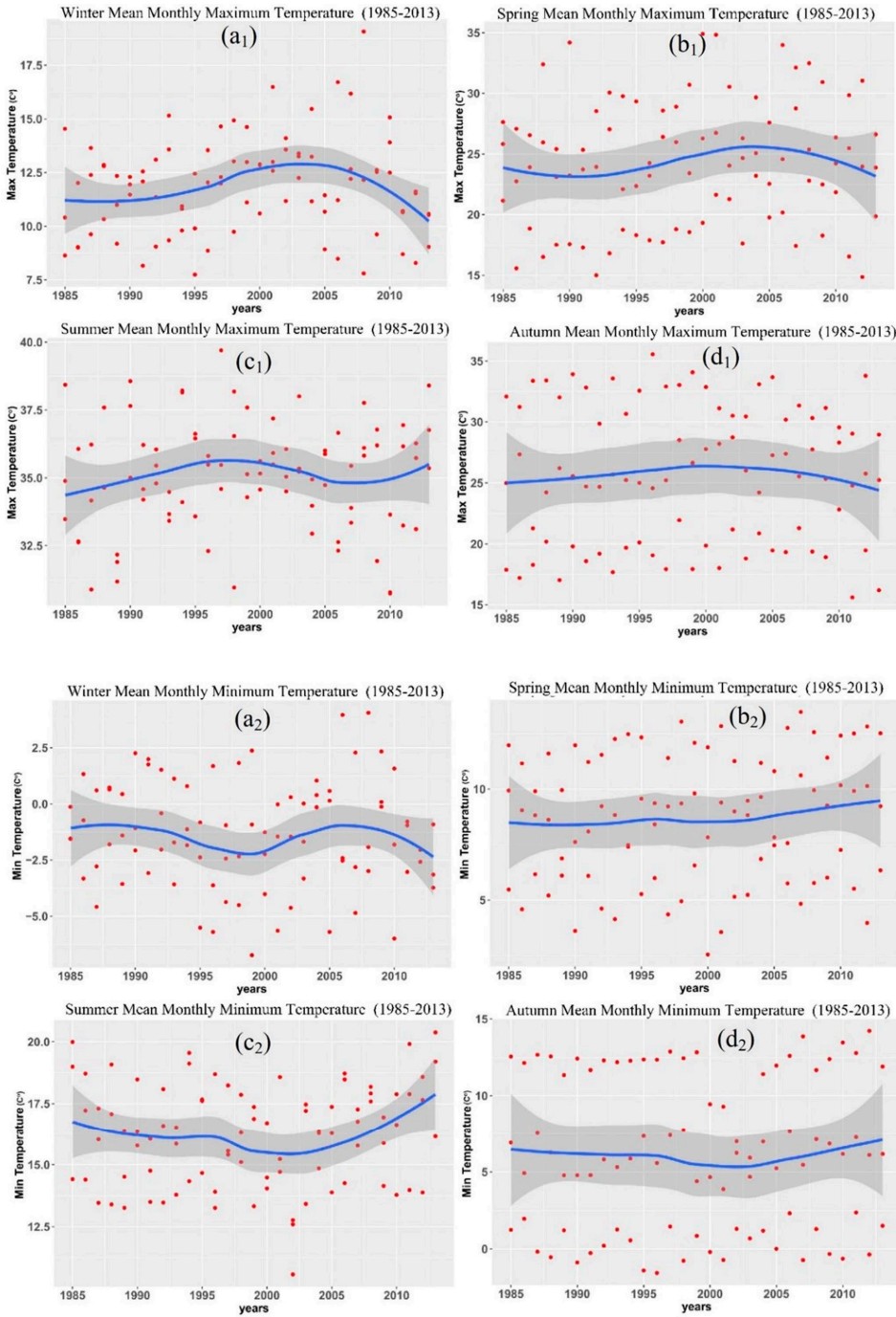

**Figure 9.** Mean monthly temperature (max and min) patterns in (°C) over the period (1985–2013) (**a1**) Winter, (**b1**) Spring, (**c1**) Summer, (**d1**) Autumn, (**a2**) Winter, (**b2**) Spring, (**c2**) Summer, (**d2**) Autumn.

**Table 8.** Summary statistics of temperature (max and min) in °C over the period (1985–2013).

| Statistics | Winter | | Spring | | Summer | | Autumn | |
|---|---|---|---|---|---|---|---|---|
| | Max | Min | Max | Min | Max | Min | Max | Min |
| Min. | 7.7 | −6.7 | 14.8 | 2.5 | 30.7 | 10.6 | 10.6 | −1.5 |
| 1st Quartile | 10.5 | −2.9 | 20 | 6.1 | 33.8 | 14.4 | 14.4 | 1.3 |
| Median | 12 | −1.5 | 24 | 9.2 | 35.2 | 16.3 | 16.3 | 5.9 |
| Mean | 11.8 | −1.4 | 24.2 | 8.7 | 35.1 | 16.2 | 16.2 | 6.1 |
| 3rd Quartile | 13 | 0.4 | 27.3 | 11.2 | 36.2 | 17.8 | 17.8 | 11.7 |

**Table 8.** *Cont.*

| Statistics | Winter | | Spring | | Summer | | Autumn | |
|---|---|---|---|---|---|---|---|---|
| | **Max** | **Min** | **Max** | **Min** | **Max** | **Min** | **Max** | **Min** |
| Max. | 19.1 | 4.1 | 34.9 | 13.5 | 39.7 | 20.4 | 20.4 | 14.2 |
| Standard Deviation | 2.0 | 2.4 | 5.1 | 2.8 | 3.5 | 2.4 | 3.0 | 5.0 |

**Table 9.** Mean annual temperature (max and min) in °C variations of baseline and GCMs under SSP245 scenario.

| Data Set | Historical 1974–1993 | | Baseline 1994–2013 | | 20's 2014–2033 | | 80's 2074–2094 | |
|---|---|---|---|---|---|---|---|---|
| | **Max** | **Min** | **Max** | **Min** | **Max** | **Min** | **Max** | **Min** |
| Observed | - | - | 23.5 | 17.8 | - | - | - | - |
| ACCESS-CM2 | 21.8 | 14.9 | 23.4 | 16.3 | 24.5 | 19.8 | 26.7 | 20.4 |
| CMCC-CM2-SR5 | 21.7 | 15.4 | 23.2 | 17.8 | 23.9 | 19.2 | 26.9 | 21 |
| GFDL-ESM4 | 22 | 16.2 | 23 | 16.9 | 24.4 | 18.2 | 26.7 | 19.3 |
| INM-CM4-8 | 22.1 | 15.6 | 22.8 | 16.3 | 23.5 | 17.5 | 27.8 | 19.5 |
| INM-CM5-0 | 22.1 | 15.7 | 22.7 | 17.2 | 24.1 | 18.6 | 26.1 | 20.9 |
| NorESM2-LM | 21.7 | 15.8 | 22.9 | 16.6 | 24.2 | 18.1 | 27.6 | 19.1 |
| NorESM2-MM | 21.7 | 15.9 | 22.7 | 17.3 | 23.9 | 17.9 | 27.7 | 20.1 |
| TaiESM1 | 21.6 | 15.5 | 23.2 | 17.5 | 24.5 | 18.5 | 26.9 | 20.7 |
| UKESM1-0-LL | 22 | 15.9 | 21.6 | 17.7 | 23.2 | 18.9 | 25.5 | 20.5 |
| GFDL-CM4 | 21.8 | 15.4 | 22.3 | 16.9 | 24.5 | 18.8 | 26.3 | 19.6 |

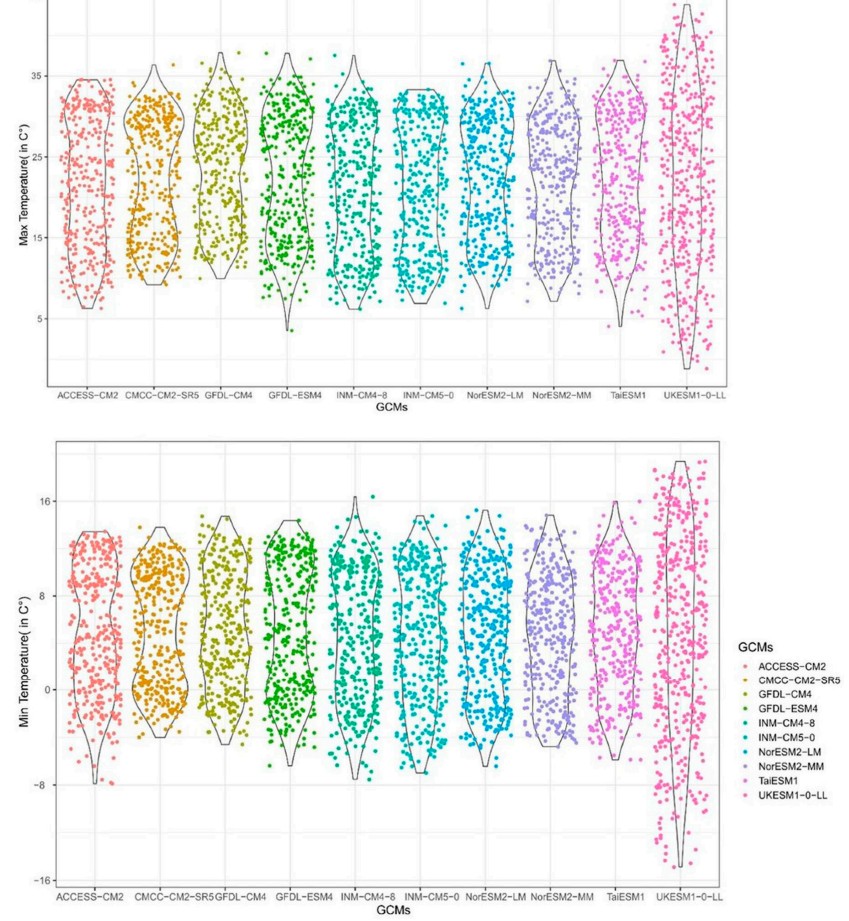

**Figure 10.** Mean monthly projected temperature (max and min) in °C spread and variation of GCMs under SSP245 scenario from 1974 to 2094.

### 3.4. Temporal Patterns of Streamflow

Streamflow has a diverse connection with climate change, as in Table 10 it can be perceived that the in-summer season streamflow reaches its peak due to warmer temperatures in the upper regions of the Indus Basin. Similarly, in the winter season, the river runoff becomes the least due to lower temperatures. Figure 11 illustrates the seasonal fluctuation in river runoff over the 1985–2013 period. The winter season has varied substantially over time; it had a rising trend until 1993 before it began to decline. The summer season has a diversified pattern, with an increasing tendency. In addition, there have been no major changes in the spring and autumn seasons, with modest increases in spring and decreases in autumn. Table 11 portrays the increasing pattern of streamflow in the future, which suggests that high temperatures can contribute to alleviating river runoff, and high temperatures can give rise to more glacier melting in the upper region of the Indus Basin. In addition, Figure 12 shows the predicted streamflow under the SSP245 scenario, which illustrates the fluctuation and dispersion of numerous models, streamflow is densely saturated between 20 to 170 $m^3$/s in almost all models. Higher peaks of more than 770 $m^3$/s are observed in the future by ACCESS-CM2 and UKESM1-0-LL. These findings emphasize the strong linkage between climate change and streamflow in the Upper Indus Basin, particularly the role of temperature in driving seasonal variations. The projected future changes have profound implications for water resource management and the need for adaptive measures to sustainably manage the region's water resources.

**Table 10.** Summary statistics of mean seasonal streamflow ($m^3$/s) over the 1985–2013 period.

| Statistics | Winter ($m^3$/s) | Spring ($m^3$/s) | Summer ($m^3$/s) | Autumn ($m^3$/s) |
|---|---|---|---|---|
| Min. | 19.4 | 19.9 | 137.8 | 31.2 |
| 1st Quartile | 30.1 | 33.6 | 258.2 | 52 |
| Median | 33.7 | 60 | 345.5 | 67.27 |
| Mean | 34.5 | 103.6 | 361.7 | 87.6 |
| 3rd Quartile | 38.27 | 164.8 | 446.7 | 105.4 |
| Max. | 55.9 | 389.4 | 738.4 | 348.6 |
| Standard Deviation | 6.7 | 112.2 | 220.6 | 85.8 |

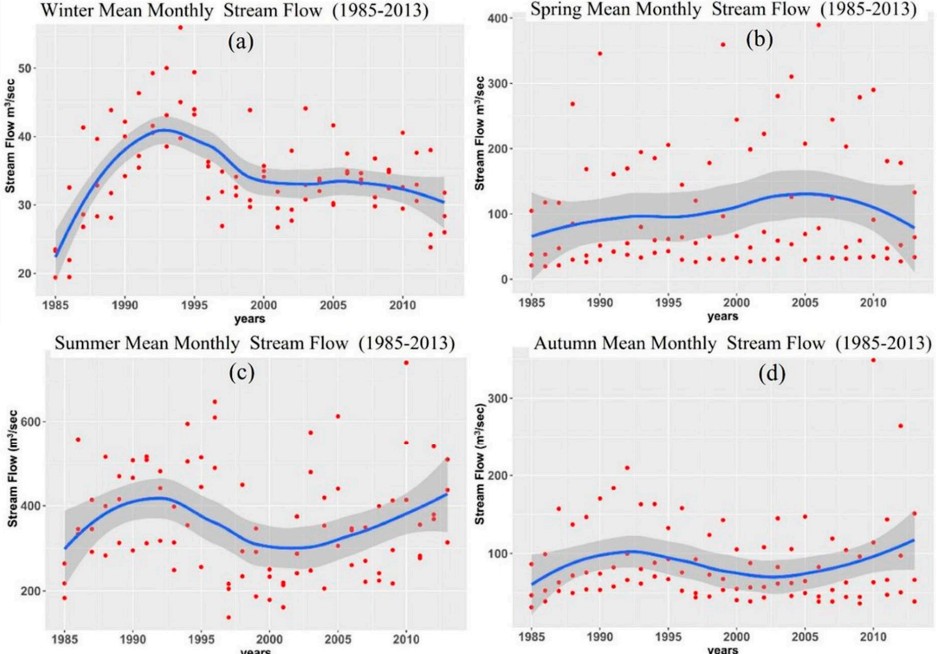

**Figure 11.** Mean monthly streamflow ($m^3$/s) patterns over the period (1985–2013) (**a**) Winter, (**b**) Spring, (**c**) Summer and (**d**) Autumn.

**Table 11.** Mean annual streamflow (m$^3$/s) variations of baseline and GCMs under SSP245 scenario.

| Data Set | Historical 1974–1993 | Baseline 1994–2013 | 20's 2014–2033 | 80's 2074–2094 |
|---|---|---|---|---|
| Observed | - | 146.5 | - | - |
| ACCESS-CM2 | 135.7 (−8%) | 147.5 | 160.2 (+9%) | 174.8 (+19%) |
| CMCC-CM2-SR5 | 133.3 (−7%) | 143.7 | 161.2 (+12%) | 177.1 (+23%) |
| GFDL-ESM4 | 134.1 (−5%) | 139.9 | 159.4 (+14%) | 181.5 (+30%) |
| INM-CM4-8 | 132.5 (−9%) | 145.5 | 156.9 (+8%) | 173.6 (+19%) |
| INM-CM5-0 | 129.9 (−12%) | 147.3 | 160.8 (+9%) | 179.4 (+22%) |
| NorESM2-LM | 135.1 (−6%) | 142.6 | 162.3 (+14%) | 183.1 (+28%) |
| NorESM2-MM | 128.5 (−12%) | 146.1 | 159.6 (+9%) | 179.9 (+23%) |
| TaiESM1 | 133.9 (−7%) | 141.3 | 163.5 (+16%) | 181.3 (+28%) |
| UKESM1-0-LL | 131.7 (−11%) | 148.3 | 160.1 (+8%) | 180.7 (+22%) |
| GFDL-CM4 | 134.6 (−6%) | 143.7 | 157.3 (+9%) | 179.6 (+25%) |

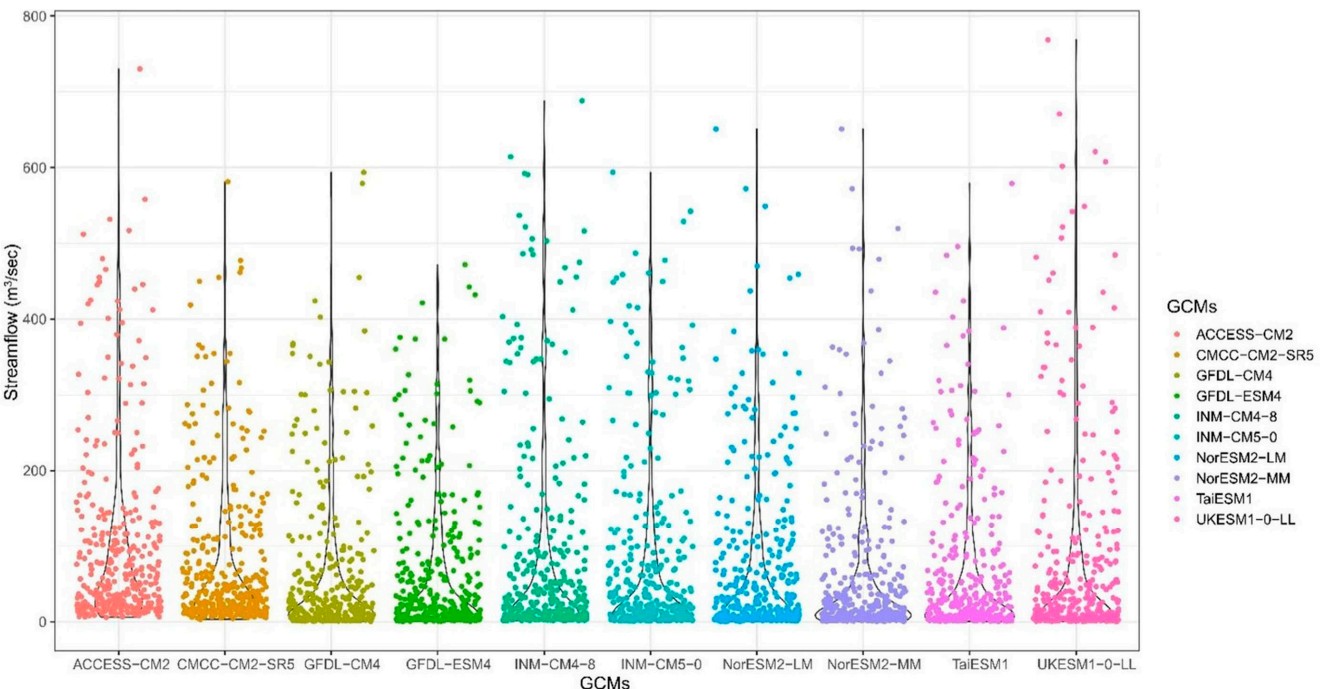

**Figure 12.** Mean monthly projected streamflow (m$^3$/s) spread and variation of GCMs under SSP245 scenario from 1974 to 2094.

## 4. Discussion

### 4.1. Climate Change Impacts on Streamflow in URIB

In a Central Himalayan watershed, predicted streamflow rose throughout the monsoon and post-monsoon seasons but dropped during the dry season, according to [55,56]. Snowmelt accounts for over 34% of total streamflow in this region, while glacier melt accounts for 26%. Both studies indicate that the Upper Indus Basin is particularly sensitive to temperature fluctuations [57,58]. In Refs. [59,60], both studies used numerous GCMs to anticipate future climatic changes from dry and cold to wet and warm. Temperature and precipitation are expected to increase, which predicts increases in annual runoff (7.57–32.12%) by 2100. The hydrological regime is primarily influenced by snow and glacier melt [61].

Khattak, Babel and Sharif [62] previously identified an uneven pattern of precipitation fluctuation in the southern upper regions of the Indus Basin (URIB) between 1967 and 2005. Bocchiola and Diolaiuti [63] confirmed comparable precipitation variations over URIB. An upward tendency in the spring and summer and a downward trend in the winter and autumn was found in this research. Precipitation is rising over time, which

is in line with [64]. Aresta, Dibenedetto and Quaranta [65] used a stochastic rainfall model for hydrological impacts, which predicted that precipitation would increase by 27% seasonally and 18% mean annual changes. According to the studies given above, we may expect a rise in precipitation in the future. Our findings align with [65,66], which underscores the sensitivity of the Upper Indus Basin to temperature fluctuations, with significant implications for streamflow. In this region, snowmelt and glacier melt contribute substantially to total streamflow, and our study reaffirms that these sources are particularly sensitive to temperature changes. Future temperature increases are anticipated in the URIB area under two alternative scenarios (A2 and A1B). Furthermore, ref. [67] utilized PRECIS RCM data to show that raising projected temperatures in the Indus Basin had comparable impacts. The hydrological regime, heavily influenced by snow and glacier melt, is set to undergo a transformation, necessitating adaptive water management strategies.

### 4.2. Importance of Watershed Management

Climate change is expected to have significant repercussions not just for the upper regions of Pakistan, but also for the downstream urban and rural areas, which rely heavily on mountain water sources for residential, agricultural, and industrial purposes. Seasonal fluctuations in precipitation and temperature may affect the agricultural calendar in the future. The hydrological regime of the upper regions of the Indus Basin is currently supplying water to the rest of the country. Changes in water quantity and quality will have disastrous effects on social and economic processes throughout the country. This study established that streamflow is very sensitive to temperature fluctuations. As the primary source of streamflow in the UIRB is snow and glacier melting, this is very harmful because it may reduce the overall glacial volume in the long term and may result in natural disasters in that area [68,69]. Water management methods may help reduce the effects of severe weather situations like floods and droughts by storing water. Agriculture, hydroelectric, industrial, and domestic uses will all take advantage of the stored water. Plantations are also advised in the region to reduce the rising trend in air temperature since streamflow is more sensitive there [70,71]. Our water comes from mountains, and we need to understand more about them. Improved scientific education and information sharing are necessary for managing water resources [31].

The Indus River is critical to Pakistan's economic growth and food production, contributing around 25% of the country's gross domestic product and providing 90% of the water used in agricultural production. According to the World Bank's study (2020–2021), a 32% decline in water by 2025 will result in about 70 million tons of food shortfall in the country. This highlights the interconnection among water–energy–food and emphasizes the need to address the pressing issue of water–energy–food in the Indus Basin. To tackle water scarcity in the Indus Basin, transboundary collaboration among basin nations is necessary to foster yearly spending of USD 10 billion by 2050. However, by adopting transboundary level collaborative policies, this cost might be decreased to a much lower USD 2 billion per year. Such collaborative measures benefit the downstream regions to enjoy lower costs of food and energy, and increased water availability, while upstream regions benefit from critical new energy investments, demonstrating the significant benefits of fostering cooperation in the region's water–energy–food nexus [72]. Our findings emphasize the importance of watershed management and water storage techniques to mitigate the impact of extreme weather events. Collaboration and information sharing, as well as the adoption of the water–energy–food nexus concept, are crucial for sustainable water resource management in the face of climate change challenges.

The "water wars" narrative has been relaunched in the past ten years, which is mainly due to climate change. The findings of the current study suggested that climate change will impact streamflow, which might increase the possibility of conflicts and disputes over the shared water resources in the region. In this context, the idea of "water diplomacy" can play a vital role in bringing sustainability to the transboundary water resources of the region. Water diplomacy can decline the potential regional water-related conflicts and

promote peace and harmony in the face of climate change. The competition for freshwater resources will increase in the region, owing to economic development, population growth, and climate change, which has increased the risk of potential "water wars" in the region. Water diplomacy is the only way to mitigate and manage these risks [73].

## 5. Conclusions

This study, centered on the Upper Indus Basin of Pakistan, has shed light on the pressing issues posed by climate change in a region crucial for water resources and the well-being of its indigenous communities. To address these concerns, we employed state-of-the-art CMIP6-based GCMs within the SWAT framework, with the primary objectives of projecting streamflow, assessing the implications of climate change on river runoff, and evaluating the changes in precipitation and temperature patterns.

The following are the study's major findings:

(1) The SWAT model delivers statistically significant outcomes during calibration ($NSE = 0.77$, $R^2 = 0.82$, $RSR = 0.48$, and $PBIAS = 13.5$) and validation ($NSE = 0.68$, $R^2 = 0.71$, $RSR = 0.57$, and $PBIAS = 14.2$).

(2) Under the SSP245 scenario, the SWAT model was used to figure out how climate change would affect streamflow. It has been found that streamflow could rise by 19–30% under different scenarios.

(3) Upper regions of the Indus Basin are very susceptible to temperature, and future changes are expected to be drastic, with maximum and minimum temperatures intensifying by almost 18%.

(4) Precipitation has an increasing pattern through the period, which might contribute very little to streamflow, and precipitation will increase by 11–17%.

(5) GCMs based on CMIP6 under SSP245 scenarios have been illustrated to be effective in predicting future climate change in the study area. A calibrated model has been used for the simulations of the (1974–1993), (1994–2013), (2014–2033), and (2074–2094) periods and all of them exhibited an increase in average annual streamflow.

**Author Contributions:** Conceptualization, K.H. and A.U.K.; data curation, K.H. and A.U.K.; formal analysis, K.H. and A.U.K.; funding acquisition, A.A.J.G. and A.U.K.; investigation, A.A.J.G.; methodology, K.H. and A.U.K.; project administration, A.A.J.G.; resources, A.A.J.G. and A.U.K.; software, K.H. and A.U.K.; supervision, A.A.J.G. and A.U.K.; validation, J.K.; visualization, J.K.; writing—original draft, K.H.; review and editing, K.H., J.K., A.U.K. and A.M.A.-A. All authors have read and agreed to the published version of the manuscript.

**Funding:** The authors are thankful to the Deanship of Scientific Research at Najran University for funding this work under the Research Group funding program grant code (NU/RG/SERC/12/47).

**Institutional Review Board Statement:** Not applicable.

**Informed Consent Statement:** Not applicable.

**Data Availability Statement:** Hydrometeorological data are available with the Pakistan Meteorological Department (PMD), Water and Power Development Authority (WAPDA), and the Irrigation Department, subject to payment of specified charges.

**Acknowledgments:** The authors are thankful to the Deanship of Scientific Research at Najran University for funding this work under the Research Group funding program grant code (NU/RG/SERC/12/47). The authors also would like to express their gratitude to the supporting staff and management of WAPDA, PMD, and the Irrigation Department for their invaluable assistance in facilitating and supplying the necessary data. Their contributions were instrumental in successfully completing this study.

**Conflicts of Interest:** The authors declare no conflict of interest.

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
