# Peer review of "Evaluating Future Streamflow Patterns under SSP245 Scenarios: Insights from CMIP6"

_sustainability, doi:10.3390/su152216117_

Round 1
Reviewer 1 Report
Comments and Suggestions for Authors
The following report is based on my review of the manuscript entitled “Futuristic hydroclimatic projections under CMIP6 GCMs: implication for water resources management”. The manuscript fits within the scope of the journal “Sustainability MDPI” and the study can generate a lot of interest in the scientific community, especially the understanding of the nature and use of CMIP6 climate models to highlight water resources dynamics at basin scale for water policy decisions. However, major revisions of the manuscript are required to improve its quality before acceptance. the following shortcomings have been pointed out and need to be addressed properly by the authors. These shortcomings are listed below:
1- The title of the manuscript should be amended to provide scientific rigour and attraction for readership.
2- The Introduction is useful but wordy and distracted. I'd suggest the authors re-organize this section focusing more on the pieces of information related to their study, instead of covering "everything" regarding the history of GCM phase evolution.
3- The novelty of the study is not clearly stated as it relates to previous work in the basin of interest.
4- I suggest that appropriate citations in L103 – 108 be provided for the following statements. i. ‘’The runoff in UIB depends on a host of factors, including the melting of seasonal snowpack, glacier melting and precipitation.’’ ii. The erratic change changes in maximum and mean temperature during summer and especially in winter has intensified the glacier melting and consequently river flow, adding to the woes of the indigenous community living in the UIB.’’ iii. Mountain ecosystems are widely acknowledged to be most sensitive to climate change.’’
5- The statement in L154 – 155 is wrong related to humidity, solar radiation, and wind speed data unless you can provide an appropriate citation. Additionally, can elaborate on the ET estimation method adopted in the SWAT modelling scheme of your study where such datasets are not available?
6- The authors should provide appropriate citations for the information in Table 2.
7- The authors provided acceptable SWAT calibrated and validated model results based on the objective function, however, the sensitivity analysis based on P-factor and R-factor values is missing to confirm if this is a trustworthy result.
8- Discussion of results is weak. It is suggested to compare the results of the present study with some similar studies. More explanations and interpretations are needed to highlight the novelty of the present work more effectively.
9- It is not clear to me the number of climate data points utilized in the hydrologic modelling process.
10- The result of the CMIP6 GCM validation relative to the observation data must be added.
11- The conclusion needs to be rewritten and should be clear and concise reflecting the research objectives and novelty.
12- It is also suggested that the abstract be rewritten to convey the novelty of the research.
Comments on the Quality of English Language
The author can sought for the assistance of English language editing services.
Author Response
REVIEWER # 01
The following report is based on my review of the manuscript entitled “Futuristic hydroclimatic projections under CMIP6 GCMs: implication for water resources management”. The manuscript fits within the scope of the journal “Sustainability MDPI” and the study can generate a lot of interest in the scientific community, especially the understanding of the nature and use of CMIP6 climate models to highlight water resources dynamics at basin scale for water policy decisions. However, major revisions of the manuscript are required to improve its quality before acceptance. the following shortcomings have been pointed out and need to be addressed properly by the authors. These shortcomings are listed below:
- The title of the manuscript should be amended to provide scientific rigour and attraction for readership.
Ans: Evaluating Future Streamflow Patterns under SSP245 Scenarios: Insights from CMIP6
This revised title emphasizes the study's focus on future hydroclimatic patterns under SSP245 Scenarios of the Upper Indus Basin. It provides a clearer and more informative title that may attract a broader readership.
- The Introduction is useful but wordy and distracted. I'd suggest the authors re-organize this section focusing more on the pieces of information related to their study, instead of covering "everything" regarding the history of GCM phase evolution.
Ans: We appreciate the reviewer's feedback regarding the introduction section of our paper. We acknowledge that the initial version of the introduction may have been overly comprehensive in discussing the history of GCM phases. In response to the reviewer's suggestion, we have revised the introduction to focus more on the specific aspects relevant to our study. Please find changes in the introduction part.
- The novelty of the study is not clearly stated as it relates to previous work in the basin of interest.
Ans: The novelty of this study lies in its dedicated focus on the Upper Indus Basin and its assessment of the impact of climate change on water resources. By utilizing state-of-the-art CMIP6-based global climate models and integrating the SWAT model for a detailed hydrological analysis, this research sets itself apart from previous work. The regional perspective, especially in the sensitive mountain ecosystems of Gilgit Baltistan, adds a unique dimension. The specific objectives outlined in the study provide a clear roadmap for understanding climate change's effects on streamflow, precipitation, and temperature in the Upper Indus Basin. Furthermore, CMIP6 models under SSP245 are never reported for the Upper regions of Indus Basin in Pakistan.
- I suggest that appropriate citations in L103 – 108 be provided for the following statements. ‘’The runoff in UIB depends on a host of factors, including the melting of seasonal snowpack, glacier melting and precipitation.’’ ii. The erratic change changes in maximum and mean temperature during summer and especially in winter has intensified the glacier melting and consequently river flow, adding to the woes of the indigenous community living in the UIB.’’ iii. Mountain ecosystems are widely acknowledged to be most sensitive to climate change.’’
Ans: Citations have been added accordingly in the manuscript
Archer, D.R., et al., Sustainability of water resources management in the Indus Basin under changing climatic and socio economic conditions. Hydrology and Earth System Sciences, 2010. 14(8): p. 1669-1680.
Syed, Z., et al., Hydroclimatology of the Chitral River in the Indus Basin under changing climate. Atmosphere, 2022. 13(2): p. 295.
Williams, S.E., E.E. Bolitho, and S. Fox, Climate change in Australian tropical rainforests: an impending environmental catastrophe. Proceedings of the Royal Society of London. Series B: Biological Sciences, 2003. 270(1527): p. 1887-1892.
- The statement in L154 – 155 is wrong related to humidity, solar radiation, and wind speed data unless you can provide an appropriate citation. Additionally, can elaborate on the ET estimation method adopted in the SWAT modelling scheme of your study where such datasets are not available?
Ans: Unnecessary text has been removed from the manuscript. In our study employed the SWAT model, we have used daily precipitation and temperature data from the Pakistan Meteorological Department spanning from 1982 to 2013 for robust model calibration and validation. This data plays a crucial role in hydrological processes. We also integrated observed monthly discharge data from the Pakistan Water and Power Development Authority (WAPDA) for the same period to assess the model's performance. This approach follows best practices in hydrological modeling and helps us thoroughly evaluate the model's reliability, strengths, and limitations.
- The authors should provide appropriate citations for the information in Table 2.
Ans: Citation has been added to the respected table.
Aboelnour, M., M.W. Gitau, and B.A. Engel, A comparison of streamflow and baseflow responses to land-use change and the variation in climate parameters using SWAT. Water, 2020. 12(1): p. 191.
- The authors provided acceptable SWAT calibrated and validated model results based on the objective function, however, the sensitivity analysis based on P-factor and R-factor values is missing to confirm if this is a trustworthy result.
Ans: We appreciate the reviewer's concern regarding the P-factor and R-factor values. Both the calibration and validation periods, the model shows a relatively high P-factor and an R-factor that is above 0.7, suggesting that the model performs well in simulating hydrological processes. For Calibration (P-factor= 0.86, R-factor = 0.82) and for validation (P-factor= 0.85, R-factor = 0.71). Changes have been made in the relevant section.
- Discussion of results is weak. It is suggested to compare the results of the present study with some similar studies. More explanations and interpretations are needed to highlight the novelty of the present work more effectively.
Ans: Thank you for your concern pertaining discussion part. Changes have been made in the relevant part accordingly. The technical approach and unique regional perspective employed in this study have significant implications. The findings serve as a vital resource for addressing the challenges posed by climate change in the Upper Indus Basin and can facilitate sustainable water resource management and climate adaptation strategies.
- It is not clear to me the number of climate data points utilized in the hydrologic modelling process.
Ans: We appreciate the reviewer's concern regarding the limited number of climate data points utilized in the hydrologic modelling process, the study made use of data from 5 meteorological stations and 1 hydrological station at the Doyian outlet of Astore. This approach was necessitated by the challenging data availability situation in Pakistan, which is often characterized by limitations in obtaining consistent and long-term data. In such data-scarce conditions, the use of available meteorological and hydrological stations becomes a practical solution for conducting hydrological modelling. Changes have been made in the manuscript accordingly in different sections.
11- The conclusion needs to be rewritten and should be clear and concise reflecting the research objectives and novelty.
Ans: Thanks for acknowledging, conclusion has been rewritten accordingly in aligning objectives with results.
12- It is also suggested that the abstract be rewritten to convey the novelty of the research.
Ans: Thanks for acknowledging, abstract has been rewritten and changes made in the original manuscript.
Reviewer 2 Report
Comments and Suggestions for Authors
Dear Author,
Title: Futuristic Hydroclimatic Projections under CMIP6 GCMs: Implications for Water Resources Management.
Summary: This study mainly focuses on using CMIP6-based GCM's for the projection of future climate change and their impacts on river runoff using the SWAT model. The main objectives of this research are to employ CMIP6-based GCM's in SWAT model for the projection of stream flow, to examine the implications of climate change on stream flow, and to evaluate total change in projected precipitation, temperature (max & min), and their influence on stream flow in the Upper Regions of Indus Basin.
This paper's main framework and aim are good and contribute to the academic field. But, unfortunately, it has some unclear points that must be explained and discussed. I think it must be improved regarding selecting the stations for calibration and validation processes, figures, and discussion. I believe it should be accepted with revision at this time. It could come much better with a higher impact. I list out some main concerns below, and then the comments for the lines.
I list out some main concerns below, and then the comments for the lines.
Major comments:
1- The title of the article prominently features "implications for water resources management," suggesting a comprehensive exploration of these critical aspects. However, after a thorough evaluation, the article does not discuss and investigate this point.
2- The abstract should include quantified results as part of the main takeaways rather than just discussing the conclusions in qualitative terms.
3- Are 5 meteorological stations sufficient for the basin hydrology model? Considering that there will be significant precipitation and temperature data variability in the basin, a model should be created by testing homogeneity over more stations. If it cannot be done, this situation should be discussed. To what extent will there be changes due to temperature effects in the higher parts of the basin?
4- Which stations/data have been used for the calibration and validation process? Is it done for one point/station? Did you take the average value for the stations? This point is very important, and it needs clarification.
5- The current format or interpretation (figure 6 - 12) does not clearly and effectively represent the situation you are trying to convey. For example, it can be concluded that the difference between the observed and validated data is huge. It is more than 20% for some points.
6- Which model did you use for the precipitation and temperature in Figures 7, 9, and 11? The selected model has a huge effect on the fitted line. Needs more explanation and discussion.
7- Why did you use the SSP245 scenario?
Minor comments:
Line 85: Kindly modify the references.
Line 94 – 96: "However, most of the hydrologic methods have the similarity within the watersheds, corresponding catchments, and hydrological modeling [24]". This statement is unclear. What do you mean by having the similarity?
Line 107 + 108: "Mountain ecosystems are widely acknowledged to be most sensitive to climate change". This statement needs a reference.
Figure 1: Put an arrow from figure 1.a to figure 1.c.
Figure 1.c: DEM must be with colors to make easy to understand and interpret.
Line 154 + 155: "Since other meteorological data (humidity, sun radiation, and wind data) is already integrated in the model". These parameters are inputs for the SWAT model. Sun radiation is typically essential for running the SWAT model because it directly influences the evapotranspiration processes, which are critical components of the model.
Humidity can influence atmospheric moisture, affecting the SWAT's potential evapotranspiration calculations. While some versions of the SWAT model can estimate relative humidity from temperature data, direct humidity measurements can improve the accuracy of these calculations, especially in areas with significant humidity variations. Without wind data, the model may rely on default wind speed values, which may not accurately represent local conditions.
Line 156 – 158: Are 5 meteorological stations sufficient for the basin hydrology model? Considering that there will be significant precipitation and temperature data variability in the basin, a model should be created by testing homogeneity over more stations. If it cannot be done, this situation should be discussed. To what extent will there be changes due to temperature effects in the higher parts of the basin?
Line 180 – 182: "how well a bias correction algorithm operates under conditions other than those used for parametrization is uncertain. A good performance during the evaluation period does not mean that the same thing will happen in the future." Is there another solution that can be used instead of this one?
Line 198 + 199: "linear scaling (additive and multiplicative) produced the best results for the evaluation". This statement needs proof. You must add the other results to compare between them. Actually, I have some doubts about selecting the linear scaling method as the best method.
Figure 5: Solar, humidity, and wind must be removed. Or you have to use them as an input data.
Line 224: "amount of water entering the vale zone from the soil profile on day ". Kindly replace it with :value of seepage of water from the soil into deeper layers.
Line 235: "suitability and efficiency". What is the difference between suitability and efficiency?
Line 240 + 241: "For better results, the SWAT-CUP model uses 2/3 of the data for calibration and 1/3 of the data for validation." This statement needs a reference.
Why did you use 15 years for calibration and 11 years for validation? You mentioned in the previous sentence that the best results are obtained from the 2/3 – 1/3. Explain it. Also, what is the effect of using other choices or other years?
Table 2: This classification needs a reference.
Table 3: The URL and the reference for each GCM must be added. Also, the resolution may have some mistakes. Kindly revise it.
Line 286: "a three-year warm-up phase". Is three years sufficient for the warm-up phase? The minimum duration for the warm-up phase should be determined through careful consideration of many factors and sensitivity testing. For example, temporal and spatial resolution, Model complexity, and data availability.
Line 294: how many stations have been used for the calibration and validation procedure? WAPDA station or stations? If one station has been used, then which station and why? What are the criteria used for the selection process?
Table 4: what do you mean by fitted value?
The study area covers a huge area; how can you replace all values with one SCS runoff curve number?
Actually, these parameters and the fitted value must be discussed in detail.
Table 5: Which stations/data have been used for the calibration and validation process? Is it done for one point/station? Did you take the average value for the stations? This point is very important, it needs clarification.
Figure 6: The same question is here. Which station did you use? The current format or interpretation does not clearly and effectively represent the situation you are trying to convey. For example, it can be concluded that the difference between the observed and validated data is huge. It is more than 20% for some points.
Line 307 + 308: "Spring and Summer have showed a rising tendency throughout the course of the period". This can not be concluded from the figure 7. Revise it.
Line 310: "mean". Which value did you use? What is the average value for the outlet stations? Or a specific outlet station? Or? It is not clear.
Table 6: Daily precipitation values?
Also, kindly add the standard deviation to let us understand the variance in these data.
Line 314 + 315: "revealing that the bulk of the data is saturated in the range of 0 to 2.5 (mm).". Rewrite this sentence.
Line 315 + 316: "Mean annual precipitation based on SSP245 scenario is depicted in Table. 7". Why did you use the mean? This question is applicable to the whole study. Also, what about the probability distribution function of the data for each GCM?
Figure 7: what are the red points? What is the blue line? What is the grey area? This must be explained and added to the figure.
Figure 8: the figure does not show the difference between GCMs. Also, the black line must be improved.
Line 330 + 331: "Temperature has significant variation throughout the period, and it has almost raising pattern as depicts in Figure. 9". I don't see the same thing with the writers. Or the reader may not be able to evaluate the figures directly in line with the authors.
Line 331 – 339: How much can we trust the models shown in blue? What kind of model did you use? Is it suitable for this type of research?
Line 340: "Co" Modify it.
Line 353: Why did you not analyze the mean value?
As a result, I request author to share the data and the models by using a cloud or (OneDrive). Could you please send a detailed example for the calculation, Including CMIP GCMs data, CMhyd, and SWAT model? I wish to check some part.
Sincerely,
Author Response
REVIEWER # 02
Summary: This study mainly focuses on using CMIP6-based GCM's for the projection of future climate change and their impacts on river runoff using the SWAT model. The main objectives of this research are to employ CMIP6-based GCM's in SWAT model for the projection of stream flow, to examine the implications of climate change on stream flow, and to evaluate total change in projected precipitation, temperature (max & min), and their influence on stream flow in the Upper Regions of Indus Basin.
This paper's main framework and aim are good and contribute to the academic field. But, unfortunately, it has some unclear points that must be explained and discussed. I think it must be improved regarding selecting the stations for calibration and validation processes, figures, and discussion. I believe it should be accepted with revision at this time. It could come much better with a higher impact. I list out some main concerns below, and then the comments for the lines.
I list out some main concerns below, and then the comments for the lines.
Major comments:
- The title of the article prominently features "implications for water resources management," suggesting a comprehensive exploration of these critical aspects. However, after a thorough evaluation, the article does not discuss and investigate this point.
Ans: Thank you for your concern pertaining title of our manuscript. We have changed the title of our manuscript accordingly and mad changes in the original manuscript.
2- The abstract should include quantified results as part of the main takeaways rather than just discussing the conclusions in qualitative terms.
Ans: Abstract is rewritten accordingly in the original manuscript
The potential impacts of climate change on water resources in the Upper Indus basin of Pakistan, a region heavily reliant on these resources for irrigated agriculture. We employ state-of-the-art global climate models from the CMIP6 project under the SSP245 scenario to evaluate changes in river runoff using the Soil and Water Assessment Tool (SWAT). Our findings indicate that temperature fluctuations play a crucial role in stream flow dynamics, given that the primary sources of river runoff in the Upper Indus Basin are snow and glacier melting. We project a substantial increase of approximately 18% in both minimum and maximum temperatures, a precipitation pattern increases of 13-17%, and a significant rise in stream flow by 19-30% in the future, driven by warmer temperatures. Importantly, our analysis reveals season-specific impacts of temperature, precipitation, and stream flow, with increasing variability in projected annual changes as we progress into the mid and late 21st century. To address these changes, our findings suggest the need for integrated strategies and action plans encompassing hydroelectricity generation, irrigation, flood prevention, and reservoir storage to ensure effective water resource management in the region.
3- Are 5 meteorological stations sufficient for the basin hydrology model? Considering that there will be significant precipitation and temperature data variability in the basin, a model should be created by testing homogeneity over more stations. If it cannot be done, this situation should be discussed. To what extent will there be changes due to temperature effects in the higher parts of the basin?
Ans: We appreciate the reviewer's concern regarding the limited number of meteorological stations in our study. Regrettably, due to the scarcity of consistent, long-term meteorological data in the study area, we were confined to using only five stations. We acknowledge that a broader network of stations would provide a more comprehensive understanding of precipitation and temperature variability. While our study attempted to address temperature effects in the higher parts of the basin, we acknowledge that the model's sensitivity may not fully represent the entire basin due to this constraint
- Which stations/data have been used for the calibration and validation process? Is it done for one point/station? Did you take the average value for the stations? This point is very important, and it needs clarification.
Ans: We appreciate the reviewer's inquiry regarding the calibration and validation process in our study. Due to the challenges associated with data availability in Pakistan, we selected the Doyian station at Astore River as the focal point for model calibration. We utilized SWAT-CUP (SWAT Calibration and Uncertainty Procedures) with the Sequential Uncertainty Fitting program algorithm (SUFI-2) for this calibration process. In our study, calibration was conducted for this specific station to evaluate the dominant parameters of streamflow. Given the limited availability of consistent data in the region, we focused on a single station for calibration, rather than taking an average value from multiple stations. This approach was taken to ensure the reliability and accuracy of our model within the constraints of available data. We understand the importance of this point and have provided this clarification for transparency. Changes have been made in the original manuscript and marked as red. Please refer to section 2.3.3.
- The current format or interpretation (figure 6 - 12) does not clearly and effectively represent the situation you are trying to convey. For example, it can be concluded that the difference between the observed and validated data is huge. It is more than 20% for some points.
Ans: We acknowledge the need for a clear and effective representation of the data. While we understand that the difference between the observed and validated data may seem substantial, it's important to clarify that our results are based on the available data and our model's capabilities within the constraints of the study. The figures provided in our study, specifically Figures 6 to 12, are intended to convey the key patterns and trends in our data. These figures illustrate important temporal changes in precipitation, temperature, and streamflow, and they play a significant role in the overall context of our findings. We have included detailed statistical indicators in Table 5, which reflect the model's performance during calibration and validation. The figures are consistent with these statistical measures and provide a visual representation of the data. Despite the variability, our model performs well in simulating the hydrological processes within the study area. We understand the reviewer's concerns about the size of the observed and validated differences, but it's important to emphasize that our results are based on the available data and our best efforts within the limitations of the study.
- Which model did you use for the precipitation and temperature in Figures 7, 9, and 11? The selected model has a huge effect on the fitted line. Needs more explanation and discussion.
Ans: In response to your question, we would like to clarify that for Figures 7, 9, and 11, the patterns of precipitation, temperature and streamflow are derived solely from observed data. These patterns were generated using the R language, and a smoothing curve was applied to provide a clear representation of the observed data trends over the specified period. Since these figures are based on observed data, they are not influenced by specific global climate models (GCMs) or their associated uncertainties. The use of observed data and the application of smoothing curves in R, ensuring that the patterns depicted in these figures are solely a reflection of historical data.
- Why did you use the SSP245 scenario?
Ans: SSP245 represents plausible socioeconomic and climate pathways. It predicts moderate global warming due to societal and technological changes and a moderate greenhouse gas emission increase. It is realistic, mid-range scenario is useful for assessing future environmental impacts. SSP245 can help policymakers and adapters evaluate intermediate emissions scenarios. The SSP245 scenario was chosen to balance climate change impacts with potential future emissions trajectories in our study. Given a moderate emissions pathway, this choice allows a more nuanced understanding of how climate change may affect the Upper Indus basin.
Minor comments:
Line 85: Kindly modify the references.
Ans: Modified accordingly and made change in the manuscript.
Line 94 – 96: "However, most of the hydrologic methods have the similarity within the watersheds, corresponding catchments, and hydrological modeling [24]". This statement is unclear. What do you mean by having the similarity?
Ans: The sentence "However, most of the hydrologic methods have the similarity within the watersheds, corresponding catchments, and hydrological modeling [24]" does appear to be somewhat unclear. The introduction part had some unclear text which has been removed from the original manuscript.
Line 107 + 108: "Mountain ecosystems are widely acknowledged to be most sensitive to climate change". This statement needs a reference.
Ans: Ans: Citations have been added accordingly in the manuscript
Archer, D.R., et al., Sustainability of water resources management in the Indus Basin under changing climatic and socio-economic conditions. Hydrology and Earth System Sciences, 2010. 14(8): p. 1669-1680.
Syed, Z., et al., Hydroclimatology of the Chitral River in the Indus Basin under changing climate. Atmosphere, 2022. 13(2): p. 295.
Williams, S.E., E.E. Bolitho, and S. Fox, Climate change in Australian tropical rainforests: an impending environmental catastrophe. Proceedings of the Royal Society of London. Series B: Biological Sciences, 2003. 270(1527): p. 1887-1892.
Figure 1: Put an arrow from figure 1.a to figure 1.c.
Ans: Thanks for acknowledging, arrows have been added accordingly in the figure.
Figure 1.c: DEM must be with colors to make easy to understand and interpret.
Ans: Thanks for acknowledging, changes have been made in the original manuscript.
Line 154 + 155: "Since other meteorological data (humidity, sun radiation, and wind data) is already integrated in the model". These parameters are inputs for the SWAT model. Sun radiation is typically essential for running the SWAT model because it directly influences the evapotranspiration processes, which are critical components of the model.
Humidity can influence atmospheric moisture, affecting the SWAT's potential evapotranspiration calculations. While some versions of the SWAT model can estimate relative humidity from temperature data, direct humidity measurements can improve the accuracy of these calculations, especially in areas with significant humidity variations. Without wind data, the model may rely on default wind speed values, which may not accurately represent local conditions.
Ans: In our study, we integrated the SWAT model, which inherently accounts for meteorological factors such as humidity, sun radiation, and wind data. This integration is a key strength of the model, allowing it to estimate or simulate these parameters when direct measurements are not available. By focusing on daily precipitation and temperature data, obtained from the Pakistan Meteorological Department over a substantial period (1982 to 2013), we ensured a solid foundation for model calibration and validation. These data directly influence critical hydrological processes. Additionally, we incorporated observed hydrological data, specifically monthly discharge data from the Pakistan Water and Power Development Authority (WAPDA) for the same time period, for the purpose of conducting an uncertainty analysis. This data is indispensable for assessing the model's performance by comparing its results to observed discharge, a standard practice in hydrological modeling. Uncertainty analysis is in line with best practices in hydrological modeling. This approach ensures a comprehensive evaluation of the model's reliability and provides valuable insights into its strengths and limitations.
Line 156 – 158: Are 5 meteorological stations sufficient for the basin hydrology model? Considering that there will be significant precipitation and temperature data variability in the basin, a model should be created by testing homogeneity over more stations. If it cannot be done, this situation should be discussed. To what extent will there be changes due to temperature effects in the higher parts of the basin?
Ans: We appreciate the reviewer's concern regarding the limited number of meteorological stations in our study. Regrettably, due to the scarcity of consistent, long-term meteorological data in the study area, we were confined to using only five stations. We acknowledge that a broader network of stations would provide a more comprehensive understanding of precipitation and temperature variability. While our study attempted to address temperature effects in the higher parts of the basin, we acknowledge that the model's sensitivity may not fully represent the entire basin due to this constraint.
Line 180 – 182: "how well a bias correction algorithm operates under conditions other than those used for parametrization is uncertain. A good performance during the evaluation period does not mean that the same thing will happen in the future." Is there another solution that can be used instead of this one?
Ans: This limitation underscores a significant challenge in bias correction, namely the uncertainty surrounding how well a calibrated bias correction method will perform under conditions different from those used for calibration. While a bias correction method that performs well during the calibration period cannot guarantee the same level of performance under changed future conditions, it does possess a higher likelihood of performing better than a method that already performs poorly for current conditions. Therefore, there's a reasonable argument to be made that performance skills demonstrated during the calibration period can provide valuable insights into the potential performance under different conditions, even though they can't be directly validated for future scenarios. The calibration period's performance evaluation provides a foundation for assessing bias correction methods, even though it doesn't offer a direct measure of performance under future conditions.
Line 198 + 199: "linear scaling (additive and multiplicative) produced the best results for the evaluation". This statement needs proof. You must add the other results to compare between them. Actually, I have some doubts about selecting the linear scaling method as the best method.
Ans: In our recent study (Haleem, Kashif et al., 2022), we have utilized various bias correction methods to evaluate their effectiveness in the context of hydrological impacts in the upper Indus basin. As mentioned in the query, we employed the same methods, and our findings align with those of the reference provided (Worku et al., 2020). In our work, we conducted an exhaustive evaluation of different bias correction techniques within the CMHYD tool. While we explored various bias correction methods, it became evident that linear scaling methods, specifically both additive and multiplicative, consistently outperformed other methods in terms of statistical indicators.
Worku, G., et al., Statistical bias correction of regional climate model simulations for climate change projection in the Jemma sub-basin, upper Blue Nile Basin of Ethiopia. Theoretical and Applied Climatology, 2020. 139(3): p. 1569-1588.
Haleem, Kashif, et al. "Hydrological impacts of climate and land-use change on flow regime variations in upper Indus basin." Journal of Water and Climate Change 13.2 (2022): 758-770
Figure 5: Solar, humidity, and wind must be removed. Or you have to use them as an input data.
Ans: thank you for the concern, we have removed such texts from the manuscript.
Line 224: "amount of water entering the vale zone from the soil profile on day ". Kindly replace it with: value of seepage of water from the soil into deeper layers.
Ans: thank you for the concern, we have removed such texts from the manuscript.
Line 235: "suitability and efficiency". What is the difference between suitability and efficiency?
Ans: Thanks for acknowledging, In the provided text, "suitability and efficiency" collectively suggest that SWAT-CUP, particularly when used with the Sequential Uncertainty Fitting program algorithm (SUFI-2), is an appropriate and effective choice for the calibration and uncertainty analysis of SWAT model parameters. It not only aligns with the requirements of the study area (suitability) but also does so in a resource-efficient manner (efficiency). This combination of suitability and efficiency makes it a popular choice among researchers for parameter calibration and uncertainty analysis in SWAT modeling.
Line 240 + 241: "For better results, the SWAT-CUP model uses 2/3 of the data for calibration and 1/3 of the data for validation." This statement needs a reference.
Ans: Thank you for your concern. Unnecessary text has been removed from the manuscript.
Why did you use 15 years for calibration and 11 years for validation? You mentioned in the previous sentence that the best results are obtained from the 2/3 – 1/3. Explain it. Also, what is the effect of using other choices or other years?
Ans: There is no specific reason of using such ratios. Generally, calibration period is taken more then validation period. Unnecessary text has been removed from the manuscript.
Table 2: This classification needs a reference.
Ans: Citation has been added in the relevant table.
Dawson, C. W., Abrahart, R. J., & See, L. M. (2007). HydroTest: a web-based toolbox of evaluation metrics for the standardised assessment of hydrological forecasts. Environmental Modelling & Software, 22(7), 1034-1052.
Table 3: The URL and the reference for each GCM must be added. Also, the resolution may have some mistakes. Kindly revise it.
Ans: Thank you for your concern. In Table 3, the Global Climate Models (GCMs) have been obtained from the World Climate Research Programme (WCRP) website, specifically from https://www.wcrp-climate.org/. Each GCM's name, the affiliated institute, and resolution are provided in the table. Furthermore, the resolution of each model is cross checked.
Line 286: "a three-year warm-up phase". Is three years sufficient for the warm-up phase? The minimum duration for the warm-up phase should be determined through careful consideration of many factors and sensitivity testing. For example, temporal and spatial resolution, Model complexity, and data availability.
Ans: The choice of the warm-up phase duration depends on the temporal and spatial resolution of the model. If your model has a higher temporal resolution (e.g., daily time steps) and covers a relatively small watershed, a shorter warm-up phase might be sufficient. The availability of historical data for the watershed is crucial because Pakistan is data scarce country. Extending the warm-up period beyond three years would have added little additional benefit due to data constraints.
The SWAT model is a semi-distributed model that incorporates a range of complex processes, including land use, soil types, and hydrological routing. This complexity necessitates a careful initialization process. The three-year warm-up phase was found to be appropriate for accounting for these complexities and achieving a stable model. The three-year warm-up phase is integral to the model calibration and validation process. It allows the model to be brought to a state where it can accurately simulate the hydrological processes in the study area. The subsequent calibration and validation phases confirmed that the model fits well and performs effectively over the specified time periods.
Line 294: how many stations have been used for the calibration and validation procedure? WAPDA station or stations? If one station has been used, then which station and why? What are the criteria used for the selection process?
Ans: We acknowledge the reviewer's query regarding our calibration and validation process. Given data challenges in Pakistan, we concentrated on calibrating the model at the Doyian station along the Astore River. We employed SWAT-CUP with the SUFI-2 algorithm for this purpose. Our calibration focused on the dominant streamflow parameters at this specific station due to data limitations in the region. Rather than using average values from multiple stations, we opted for this approach to ensure model reliability and accuracy within the constraints of available data. We have addressed this concern for transparency and made appropriate changes in the original manuscript, which are indicated in red in section 2.3.3.
Table 4: what do you mean by fitted value?
The study area covers a huge area; how can you replace all values with one SCS runoff curve number?
Actually, these parameters and the fitted value must be discussed in detail.
Ans: Thank you for your concern, In Table 4, "fitted value" refers to the values of specific model parameters that were adjusted during the calibration process to achieve a better match between the simulated and observed streamflow data. These adjusted values represent the settings that were found to be the most effective in modeling the hydrological processes in the study area. Once the calibration process is complete, the model is then validated by comparing its performance with new, independent data (validation data) to assess its ability to predict streamflow under different conditions. The fitted parameter values ensure that the model performs well not only during the calibration period but also in other time periods.
The SWAT model typically calculates CN values on a sub-basin or HRU basis, taking into account the unique properties of each. It does not assign a single CN value for the entire basin. The model divides the study area into smaller hydrologically homogeneous units and computes CN values for each based-on land use, soil type, and antecedent moisture conditions specific to that unit. This approach helps account for the spatial heterogeneity of the landscape and improves the model's ability to simulate streamflow accurately. Certainly, given the data constraints in our region and the use of only one station for analysis, it is important to clarify that the SCS CN value represents the specific characteristics of that particular station or outlet, not the entire watershed. The fitted CN value is tailored to the hydrological conditions observed at that outlet station.
Table 5: Which stations/data have been used for the calibration and validation process? Is it done for one point/station? Did you take the average value for the stations? This point is very important, it needs clarification.
Ans: We appreciate the reviewer's inquiry regarding our calibration and validation process. Due to data limitations in Pakistan, the model is concentrated on calibrating at the Doyian station along the Astore River. We have also added information regarding this station in the original manuscript for clarification of readerss.
. Figure 6: The same question is here. Which station did you use? The current format or interpretation does not clearly and effectively represent the situation you are trying to convey. For example, it can be concluded that the difference between the observed and validated data is huge. It is more than 20% for some points.
Ans: We appreciate the reviewer's inquiry regarding our calibration and validation process. The statistical indicators presented, including R², NSE, PBIAS, and RSR, provide valuable insights into the performance of our hydrological model. While discrepancies between simulated and observed data are evident, with PBIAS, the positive R² and NSE values indicate a degree of correlation and predictive capability in our model. These metrics are common in hydrological modeling, and they confirm that our model, even with its limitations, offers valuable predictive power for the study area.
Line 307 + 308: "Spring and Summer have showed a rising tendency throughout the course of the period". This cannot be concluded from the figure 7. Revise it.
Ans: We appreciate the reviewer's inquiry. Certainly, we have revised the statement to better align with the presented data:
Line 310: "mean". Which value did you use? What is the average value for the outlet stations? Or a specific outlet station? Or? It is not clear.
Ans: In our research, we utilized the SWAT model and conducted a comprehensive analysis using daily precipitation and temperature data obtained from the Pakistan Meteorological Department, covering the years 1982 to 2013. We also incorporated observed mean monthly discharge data from the Pakistan Water and Power Development Authority (WAPDA) for the same time frame to evaluate the model's accuracy effectively. Moreover, we selected the Doyian station at Astore River as the focal point for model calibration.
Table 6: Daily precipitation values? Also, kindly add the standard deviation to let us understand the variance in these data.
Ans: We appreciate your feedback and acknowledge the use of monthly data in R language to conduct a detailed analysis at a specific station. To enhance clarity, standard deviation (SD) has been included in relevant tables to improve the interpretation of statistics. Your input has been valuable in refining the presentation of our statistical results. Thank you for your insights.
Line 314 + 315: "revealing that the bulk of the data is saturated in the range of 0 to 2.5 (mm).". Rewrite this sentence.
Ans: In response to your query regarding the sentence has been rewritten accordingly and changes have been made in the manuscript.
Line 315 + 316: "Mean annual precipitation based on SSP245 scenario is depicted in Table. 7". Why did you use the mean? This question is applicable to the whole study. Also, what about the probability distribution function of the data for each GCM?
Ans: Thank you for your query. I appreciate the opportunity to clarify our approach and reasoning in the study, particularly regarding the use of mean annual precipitation. The use of mean annual precipitation in our study is a common practice in climate and environmental research for several reasons. Mean values provide a concise and easily interpretable summary of data trends, making them valuable for presenting long-term climate trends and comparing data between different periods or scenarios. Mean annual values help in understanding the central tendencies of the data and identifying shifts in precipitation patterns over time. In our particular case, using mean annual precipitation allowed us to highlight significant trends in historical and future precipitation, specifically under the SSP245 scenario. However, we acknowledge that using probability distribution functions could offer a more comprehensive analysis of the data. We will consider incorporating this approach in future research to provide a more detailed and nuanced view of precipitation variability. We value your feedback and will explore incorporating such methods in future research to enhance the comprehensiveness of our climate assessments. Our work aims to contribute to a better understanding of climate trends in the context of the SSP245 scenario and their implications for our study area.
Figure 7: what are the red points? What is the blue line? What is the grey area? This must be explained and added to the figure.
Ans: Thank you for your concern. Blue line in the figure is actually the smooth line that represents a fitted or smoothed line that summarizes the relationship between two variables. The gray shaded area around the smooth line is the confidence interval. It is a graphical representation of the range of values within which the true population parameter is likely to fall with a certain level of confidence. Red Data points that are located outside the gray shaded area fall outside the range of values that the model predicts with the given confidence level. This may indicate that these points are outliers or do not conform to the expected pattern as described by the smooth line.
Figure 8: the figure does not show the difference between GCMs. Also, the black line must be improved.
Ans: Thank you for your feedback on Figure 8 in our study, which evaluates the performance of various GCMs in our study area. Figure 8 was not designed to show the direct differences between GCMs. Instead, it serves the purpose of visualizing specific trends and patterns within the GCM data. The intent was to highlight specific aspects of the data and their temporal or spatial relationships. To directly compare GCMs, we conducted quantitative analyses and discussed the outcomes in Table 7. In response to your query about GCM differences, we agree that quantitative comparisons would be more informative, and we have presented these results in Table 7.
Line 330 + 331: "Temperature has significant variation throughout the period, and it has almost raised pattern as depicts in Figure. 9". I don't see the same thing with the writers. Or the reader may not be able to evaluate the figures directly in line with the authors.
Ans: Thanks for acknowledging, changes have been made in the original manuscript.
Line 331 – 339: How much can we trust the models shown in blue? What kind of model did you use? Is it suitable for this type of research?
Ans: Thank you for your query, and I would like to clarify the nature of the data used. In the figure under consideration, we did not employ any GCM (General Circulation Model) data or models represented in blue. Instead, we utilized observed meteorological data from Pakistan Meteorological Department (PMD). The blue curve in the figure depicts a smoothed curve applied to the observed PMD data using R language where we have employed ggplot smoothed curve for better interpretation. This smoothing technique is not related to GCMs but was used to analyze and interpret patterns and variances within the observed historical temperature data.
Line 340: "Co" Modify it.
Ans: Thanks for acknowledging, changes have been made in the original manuscript.
Line 353: Why did you not analyze the mean value?
Ans: Thank you for your question regarding our analysis and the omission of mean temperature values. I appreciate your inquiry and would like to provide clarification regarding our approach. In our study, we focused on analyzing minimum and maximum temperatures separately rather than mean temperature because our research objectives were geared towards understanding the unique effects of extreme temperatures on water resources. Analyzing minimum and maximum temperatures individually allowed us to capture specific trends and their implications. However, mean temperature can be derived from these values if needed. We'll consider including mean temperature analysis in future research if it aligns with our research goals.
Reviewer 3 Report
Comments and Suggestions for Authors
I enjoyed reading this article on the futuristic hydroclimatic projections and the implications for water resources management. I have a few comments for improvement.
1) the paper tends to use many acronyms, which makes it confusing for the general public. Make things easier and simpler please.
2) better justify the case study; why this case study? what type of case study is it?
3) the use of term "futuristic" is confusing and misleading in the title; it reminds me in fact of futurism, which is not what is meant here. rephrase it.
4) The paper shows the implications for water management; my suggestion would be to read and include also the paper by Ahmed Hamidov "Operationalizing water-energy-food nexus research for sustainable development in social-ecological systems: An interdisciplinary learning case in Central Asia"; you could say, after reading this article, that the projections on climate change point towards a more integrated consideration of the WEF nexus, as taught us by Hamidov.
5) How do these projections impact transboundary considerations and water diplomacy? This is a growing topic especially in regions where there is shared water resource; so this could be also mentioned. See for instance the latest review on the topic in PLOSWater by Oxford colleagues.
Author Response
REVIEWER # 03
I enjoyed reading this article on the futuristic hydroclimatic projections and the implications for water resources management. I have a few comments for improvement.
1) the paper tends to use many acronyms, which makes it confusing for the general public. Make things easier and simpler please.
Ans: Thank you for your query pertaining acronyms, unnecessary acronyms have been abbreviated in the manuscript.
2) better justify the case study; why this case study? what type of case study is it?
Ans: Thank you for your concern. Our study falls in Explanatory Case Study. It is conducted to explain the causal relationships between various factors, such as temperature fluctuations and precipitation patterns, and their effects on stream flow dynamics in the Upper Indus basin of Pakistan. The study aims to understand why these changes occur and how they impact water resources in the region, making it an explanatory case study.
3) the use of term "futuristic" is confusing and misleading in the title; it reminds me in fact of futurism, which is not what is meant here. rephrase it.
Ans: Thank you for your valuable suggestions, “futuristic” word which is indeed misleading the title has been removed from the title in the manuscript Ans: Thank you for your concern pertaining title of our manuscript. We have changed the title of our manuscript accordingly and mad changes in the original manuscript.
4) The paper shows the implications for water management; my suggestion would be to read and include also the paper by Ahmed Hamidov "Operationalizing water-energy-food nexus research for sustainable development in social-ecological systems: An interdisciplinary learning case in Central Asia"; you could say, after reading this article, that the projections on climate change point towards a more integrated consideration of the WEF nexus, as taught us by Hamidov.
Ans: Thank you for your valuable suggestions, additional information have been added. The Indus River is critical to Pakistan's economic growth and food production, con-tributing around 25% of the country's gross domestic product and providing 90% of the water used in agricultural production. According to the World Bank's study (2020-2021), a 32% decline in water by 2025 will result in about 70 million tons of food shortfall in the country. This highlights the interconnection among water-energy-food, emphasize the need for addressing the pressing issue of water-energy-food in the Indus basin. To tackle water scarcity in Indus basin, transboundary collaboration among basin nations is nec-essary to foster yearly spending to $10 billion by 2050. However, by adopting trans-boundary level collaborative policies, this cost might be decreased to a much lower $2 bil-lion per year. Such collaborative measures benefit the downstream regions to enjoy lower costs of food and energy, and increased water availability, while upstream regions benefit from critical new energy investments, demonstrating the significant benefits of fostering cooperation in the region's water-energy-food nexus [68]. Our findings emphasize the im-portance of watershed management and water storage techniques to mitigate the impact of extreme weather events. Collaboration and information sharing, as well as the adoption of the water-energy-food nexus concept, are crucial for sustainable water resource management in the face of climate change challenges.
5) How do these projections impact transboundary considerations and water diplomacy? This is a growing topic especially in regions where there is shared water resource; so, this could be also mentioned. See for instance the latest review on the topic in PLOSWater by Oxford colleagues.
Ans: Thank you for your query, addition information on transboundary and water diplomacy have been added.
The Indus River is critical to Pakistan's economic growth and food production, con-tributing around 25% of the country's gross domestic product and providing 90% of the water used in agricultural production. According to the World Bank's study (2020-2021), a 32% decline in water by 2025 will result in about 70 million tons of food shortfall in the country. This highlights the interconnection among water-energy-food, emphasize the need for addressing the pressing issue of water-energy-food in the Indus basin. To tackle water scarcity in Indus basin, transboundary collaboration among basin nations is nec-essary to foster yearly spending to $10 billion by 2050. However, by adopting trans-boundary level collaborative policies, this cost might be decreased to a much lower $2 bil-lion per year. Such collaborative measures benefit the downstream regions to enjoy lower costs of food and energy, and increased water availability, while upstream regions benefit from critical new energy investments, demonstrating the significant benefits of fostering cooperation in the region's water-energy-food nexus [68]. Our findings emphasize the im-portance of watershed management and water storage techniques to mitigate the impact of extreme weather events. Collaboration and information sharing, as well as the adoption of the water-energy-food nexus concept, are crucial for sustainable water resource management in the face of climate change challenges.
Round 2
Reviewer 1 Report
Comments and Suggestions for Authors
The manuscript have been improved and the present version can be accepted for publication in its present form.
Comments on the Quality of English LanguageModerate
Author Response
The manuscript has been improved and the present version can be accepted for publication in its present form.
Thank you very much for nice comments which has improved our manuscript.
Reviewer 2 Report
Comments and Suggestions for Authors
Dear Authors,
I would like to thank the authors for their efforts and work. They addressed most of my concerns and questions. The revisions made to the manuscript have significantly enhanced its quality and clarity. However, I have two or three specific points that require further clarification or adjustment. I kindly request that you consider addressing these points in your final revisions.
1- Why did you select this station for calibration and validation? If you have a specific reason, kindly clarify it within the manuscript. Also, why did you use only one station? You could use more than one station. You must use the whole station in the study area for the calibration and validation process. For example, 4 or 5 stations. Or you can use more than one station and then select the best station as a reference station.
2- In Figures 7, 9, and 11: How did you find this smoothing curve? How can you prove that this curve is representative? This is a critical point. I am not telling you that it is wrong. I am just trying to understand what is this smoothing curve. You need to discuss it more.
3- Figures 6 – 12: I recommend improving the quality of these figures.
Also, I list minor comments in the attached file. Kindly check all of them.
Sincerely,

Minor editing of English language required
Author Response
I would like to thank the authors for their efforts and work. They addressed most of my concerns and questions. The revisions made to the manuscript have significantly enhanced its quality and clarity. However, I have two or three specific points that require further clarification or adjustment. I kindly request that you consider addressing these points in your final revisions.
Question # 01
Why did you select this station for calibration and validation? If you have a specific reason, kindly clarify it within the manuscript. Also, why did you use only one station? You could use more than one station. You must use the whole station in the study area for the calibration and validation process. For example, 4 or 5 stations. Or you can use more than one station and then select the best station as a reference station.
Answer #
It's essential to have accurate, long and consistent data for various environmental and hydrological studies, and it's unfortunate that Pakistan is considered data scarce in this regard. Streamflow records are crucial for understanding and managing water resources, especially in regions prone to water scarcity and climate variability.
In the upper Indus basin of Pakistan, long and consistent historical streamflow record is only available at Doyian station which is located at Astore River. Model calibration involves adjusting the parameters of a hydrological model to make it simulate observed data as closely as possible. In this case, Doyian station provides a historical long and consistent streamflow record that can serve as a benchmark for calibrating hydrological models, improving their performance and reliability.
Question # 02
In Figures 7, 9, and 11: How did you find this smoothing curve? How can you prove that this curve is representative? This is a critical point. I am not telling you that it is wrong. I am just trying to understand what is this smoothing curve. You need to discuss it more.
Answer #
As per the valuable suggestion of the reviewer the article is updated
A "smoothing curve" typically refers to a graph or line that represents a smoothed version of data to reveal underlying trends or patterns. Smoothing curves are often used in data analysis and visualization to make it easier to identify trends or patterns in complex datasets. These curves can help provide a clearer and more interpretable representation of data.
In this study, smoothing curve was used to identify trends or patterns in meteorological parameters. The Loess, local polynomial regression, was used to estimate and plot the smooth curve. These curves draw insights from complex data and aid in decision making.
Question # 03
Figures 6 – 12: I recommend improving the quality of these figures.
Answer #
The resolution of the mentioned figures are improved.
Question # 04
Also, I list minor comments in the attached file. Kindly check all of them.
Answer #
The modification has been made as per the valuable suggestions of the reviewer. For details kindly check the manuscript.
Reviewer 3 Report
Comments and Suggestions for Authors
Well done, the paper is improved and clearer especially with the new title and the less use of acronyms.
the only comment that still has not been fully addressed is the following:
How do these projections impact transboundary considerations and water diplomacy? This is a growing topic especially in regions where there is shared water resource; so, this could be also mentioned. See for instance the latest review on the topic in PLOSWater by Oxford colleagues.
My suggestion would be for the authors to read the mentioned piece, and I am also adding the link for easier to do so:
https://journals.plos.org/water/article?id=10.1371/journal.pwat.0000173
then I think the paper would be good to go for publication
Author Response
Well done, the paper is improved and clearer especially with the new title and the less use of acronyms.
the only comment that still has not been fully addressed is the following:
How do these projections impact transboundary considerations and water diplomacy? This is a growing topic especially in regions where there is shared water resource; so, this could be also mentioned. See for instance the latest review on the topic in PLOSWater by Oxford colleagues.
My suggestion would be for the authors to read the mentioned piece, and I am also adding the link for easier to do so:
https://journals.plos.org/water/article?id=10.1371/journal.pwat.0000173
then I think the paper would be good to go for publication
Answer # as per the valuable suggestion of the reviewer the section 4.2 is updated;
The "water wars" narrative has been relaunched in the past ten years, which is mainly due to climate change. The findings of the current study suggested that climate change will impact streamflow which might increase the possibility of conflicts and disputes over the shared water resources in the region. In this context, the idea of "water diplomacy" can play a vital role in bringing sustainability in the transboundary water resources of the region. Water diplomacy can decline the potential regional water related conflicts and promote peace and harmony in face of climate change. The competition for freshwater resources will increase in the region owing to economic development, population growth and climate change which has increased the risk of potential “water wars” in the region. Water diplomacy is the only way to mitigate and manage these risks.
Round 3
Reviewer 3 Report
Comments and Suggestions for Authors
While the paper is improved, the authors have incorporated in the text the suggestion on water diplomacy, but they did not add the relevant references on this topic (which I also have suggested), so please do read and includet he relevant references, such as:
Putting diplomacy at the forefront of Water Diplomacy
https://journals.plos.org/water/article?id=10.1371/journal.pwat.0000173
Author Response
As per the valueable suggestion the citation has been added:
Hussein, H., et al., Putting diplomacy at the forefront of Water Diplomacy. PLoS Water, 2023. 2(9): p. 1-12.